# Quantum trajectory framework for general time-local master equations

Brecht Donvil [1,2✉] & Paolo Muratore-Ginanneschi [1✉]

Master equations are one of the main avenues to study open quantum systems. When the master equation is of the Lindblad–Gorini–Kossakowski–Sudarshan form, its solution can be "unraveled in quantum trajectories" i.e., represented as an average over the realizations of a Markov process in the Hilbert space of the system. Quantum trajectories of this type are both an element of quantum measurement theory as well as a numerical tool for systems in large Hilbert spaces. We prove that general time-local and trace-preserving master equations also admit an unraveling in terms of a Markov process in the Hilbert space of the system. The crucial ingredient is to weigh averages by a probability pseudo-measure which we call the "influence martingale". The influence martingale satisfies a 1*d* stochastic differential equation enslaved to the ones governing the quantum trajectories. We thus extend the existing theory without increasing the computational complexity.

[1] University of Helsinki, Department of Mathematics and Statistics, P.O. Box 68, FIN-00014 Helsinki, Finland. [2] Present address: Institute for Complex Quantum Systems and IQST, Ulm University, Albert-Einstein-Allee 11, D-89069 Ulm, Germany. ✉email: brecht.donvil@uni-ulm.de; paolo.muratore-ginanneschi@helsinki.fi

Actual quantum systems are open: they unavoidably interact, even if slightly, with their surrounding environment[1]. A useful phenomenological approach is to conceptualize the interaction as a generalized measurement performed by the environment onto the system[2,3]. As a consequence, the state vector of an open system follows stochastic trajectories in the Hilbert space of its stylized isolated counterpart. These trajectories are characterized by sudden transitions, quantum jumps. Since the experimental breakthroughs[4,5] quantum jumps have been observed in atomic and solid-state single quantum systems under indirect measurement see e.g., ref. [1] for an overview. Quantum trajectory theory[6–12] (see also refs. [2,3] for textbook presentations) connects these experimental results to the axiomatic theory of continuous measurement[13,14] which derives deterministic master equations governing the dynamics of a quantum system open also in consequence of the perturbation due to measurement. According to quantum trajectory theory, the state operator of any open system whose evolution is governed by the Lindblad–Gorini–Kossakowski–Sudarshan equation[2,3,15] can be "unraveled" i.e., represented as a statistical average over random realizations of post-measurement states. Mathematically, the average is computed over the realizations of a stochastic process describing the effect on the system of the interaction with an environment subject to continuous monitoring by a measurement device[2,3,7,10,16]. The precise definition of the stochastic process is contextual to the measurement scheme. Different measurement schemes result in distinct unravelings. In all cases, the stochastic process' evolution law subsumes unitary dynamics with random collapses of the state vector occurring in consequence of an indirect measurement whilst continuously preserving the system's Bloch hyper-sphere. Finally, in order to permit a measurement interpretation, the stochastic process must be non-anticipating: the statistics up to the present observation must be invariant with respect to future measurement events[17,18]. The theory of quantum trajectories is still under active development, see e.g., refs. [19,20]. Recent experiments even support the possibility of using the theory to identify precursors of the imminent occurrence of a jump[21].

Time-local and trace-preserving master equations encountered in applications belong to a class larger than that specified by the Lindblad–Gorini–Kossakowski–Sudarshan form. This more general class consists of master equations generating completely bounded maps. It is well known (see e.g., ref. [22]) that the fundamental solution of a Lindblad–Gorini–Kossakowski–Sudarshan master equation is completely positive: it maps any positive operator on the system's Hilbert space, and eventually any extension of it by the tensor product with the identity map on an auxiliary arbitrarily sized Hilbert space, into a positive operator. Completely bounded maps are those in the larger class defined by weakening the positivity requirement into that of boundedness[23].

Master equations with completely bounded fundamental solution are typically obtained by time convolutionless perturbation theory[24] or by tracing environment degrees of freedom in Gaussian system-environment models of Bosons e.g., refs. [25,26] or Fermions e.g., refs. [27,28], and in other exactly integrable models of system-environment interactions e.g., ref. [29]. We refer the reader to[30] for a discussion of the reasons why master equations generating completely bounded maps are considered in applications, including a phenomenological discussion of the domain of validity. More theoretical arguments upholding the physical relevance of completely bounded maps directly from the postulates of quantum mechanics are presented in refs. [31–35] see, however, also ref. [36].

The importance of extending quantum trajectory theory to master equations generating completely bounded maps has been recognized early in the literature see e.g., § 9 of ref. [2]. Existing frameworks for unraveling, however, require either the introduction of ancillary Hilbert spaces[37] or to postulate memory and prescience effects between trajectories[38,39] without a measurement interpretation[40,41].

Here, we prove that master equations generating completely bounded maps admit an unraveling in non-anticipating quantum trajectories on the system's Bloch hyper-sphere as it is the case under the more restrictive hypothesis of complete positivity. Specifically, we show that the quantum trajectories are realizations of the solutions of a system of ordinary stochastic differential equations driven by counting processes. The only requirement is that each realization of a quantum trajectory enters the Monte-Carlo average with its own weight factor. The weight factor is a martingale $\mu_t$, a stochastic process whose expectation value is conserved on average (see e.g., ref. [42]) so to ensure trace preservation. At any time $t$, the martingale $\mu_t$ obeys on its turn an ordinary stochastic differential equation enslaved to those governing the state vector of the system. For reasons that will become clear, we refer to $\mu_t$ as the "influence martingale".

We illustrate our main result in integrable models whose unravelings in the Hilbert space of the system was previously believed to hinge upon memory and prescience effects or simply not possible because of non-positive preserving dynamics (Redfield equation). Besides a measurement interpretation, unravelings provide a numerical avenue to integrate open quantum systems in high dimensional Hilbert spaces[9]. In particular, it is well known (see e.g., chapter 7 of ref. [2]) that the integration times of the state operator respectively computed from the master equation and from an average over quantum trajectories is expected to undergo a cross-over as the dimension of the Hilbert space increases. We verify the existence of the cross-over in a test case using QuTiP[43], a widely applied toolbox for efficient numerical simulations of open quantum systems. We emphasize that the convergence of unravelings based on the influence martingale is guaranteed by the well-established theory of ordinary stochastic differential equations with jumps see e.g., ref. [44].

Finally, we show how the influence martingale naturally accounts for photo-current oscillations which are observed in experimental quantum optics.

## Results

Given a microscopic unitary dynamics, a partial trace implemented for example with the help of time convolutionless perturbation theory[24] (see also e.g., chapters 9 and 10 of ref. [2]) yields

$$\dot{\boldsymbol{\rho}}_t = -\imath \left[\mathrm{H}_t, \boldsymbol{\rho}_t\right] + \sum_{\ell=1}^{\mathscr{L}} \Gamma_{\ell,t} \frac{\left[\mathrm{L}_\ell, \boldsymbol{\rho}_t \mathrm{L}_\ell^\dagger\right] + \left[\mathrm{L}_\ell \boldsymbol{\rho}_t, \mathrm{L}_\ell^\dagger\right]}{2} \quad (1)$$

The master equation (1) embodies the universal form of a time-local and trace-preserving evolution law.

In (1), the Hamiltonian $\mathrm{H}_t$ is the generator of a unitary dynamics. The collection $\{\mathrm{L}_\ell\}_{\ell=1}^{\mathscr{L}}$ consists of so-called Lindblad operators modeling the interaction with the environment. The weights $\Gamma_{\ell,t}$'s are related to the probability per unit of time of the collapse associated to the Lindblad operators they couple to (1). Khalfin's theorem[45] forbids exponential decay in quantum mechanics outside the intermediate asymptotic singled out by the weak coupling scaling limit[14,46]. Thus the $\Gamma_{\ell,t}$'s are in general time dependent functions with arbitrary sign. We only require them to be bounded. The celebrated Lindblad-Gorini-Kossakowski-Sudarshan master equation is thus a special case of (1) corresponding to the complete positivity conditions

$$\Gamma_{\ell,t} \geq 0 \quad \ell = 1, \ldots, \mathscr{L}. \quad (2)$$

The conditions (2) are usually derived from microscopic models

in the weak coupling limit[14]. We emphasize that even when the conditions (2) do not hold, (1) may still admit completely positive solutions but only for special initial states and initial times. These cases correspond to completely positive but not completely positive divisible dynamical maps[22,47].

The gist of the proof of the unraveling of (1) via the influence martingale is based on an extension of Girsanov's change of measure formula, a well known result in the theory of stochastic processes (see e.g., ref. [42]). Roughly speaking, Girsanov formula expresses the average of a generic functional $F_t$ up to time $t$ of a stochastic process in terms of the weighted average of the same functional now evaluated over a second distinct stochastic process

$$\tilde{E}(F_t) = E(M_t F_t). \tag{3}$$

Here $E$ and $\tilde{E}$ denote the expectation values with respect to the probability measures of the two stochastic processes. Girsanov's theorem states that the scalar weighing factor $M_t$ must be a positive definite martingale[42] satisfying for all $t$

$$E(M_t) = 1. \tag{4}$$

The extension we propose consists in relinquishing the requirement that the martingale be positive definite. We will return below on the interpretation of the negative values of the influence martingale. We now turn to detail the proof of the unraveling.

Our aim is to prove that a state operator solution of (1) always admits the representation

$$\boldsymbol{\rho}_t = E\left(\mu_t \boldsymbol{\psi}_t \boldsymbol{\psi}_t^\dagger\right). \tag{5}$$

Here $E$ denotes the expectation value operation, $\boldsymbol{\psi}_t$ is a stochastic state vector at time $t$ defined in the Hilbert space of the system and $\boldsymbol{\psi}_t^\dagger$ is its adjoint dual. Finally, $\mu_t$ is a scalar stochastic process enjoying the martingale property. We prescribe $\boldsymbol{\psi}_t$, $\boldsymbol{\psi}_t^\dagger$ and $\mu_t$ to obey evolution laws such that the expectation value (5) indeed solves (1).

First, we require that the state vector solve the Itô stochastic differential equation[9]

$$d\boldsymbol{\psi}_t = -\imath H_t \boldsymbol{\psi}_t dt - \sum_{\ell=1}^{\mathscr{L}} \Gamma_{\ell,t} \frac{L_\ell^\dagger L_\ell - \left\|L_\ell \boldsymbol{\psi}_t\right\|^2}{2} \boldsymbol{\psi}_t dt \\ + \sum_{\ell=1}^{\mathscr{L}} d\nu_{\ell,t}\left(\frac{L_\ell \boldsymbol{\psi}_t}{\left\|L_\ell \boldsymbol{\psi}_t\right\|} - \boldsymbol{\psi}_t\right) \tag{6}$$

In (6), the $\{\nu_{\ell,t}\}_{\ell=1}^{\mathscr{L}}$ are a collection of counting processes (see e.g., refs. [2,3,7,16]). The statistics of the counting process increments $d\nu_{\ell,t}$'s are fully specified for $\ell, k = 1, \ldots, \mathscr{L}$ by

$$d\nu_{\ell,t} d\nu_{k,t} = \delta_{\ell,k} d\nu_{\ell,t} \tag{7}$$

$$E(d\nu_{\ell,t}|\boldsymbol{\psi}_t, \boldsymbol{\psi}_t^\dagger) = \imath_{\ell,t}\left\|L_\ell \boldsymbol{\psi}_t\right\|^2 dt \tag{8}$$

where $\{\imath_{\ell,t}\}_{\ell=1}^{\mathscr{L}}$ is a collection of strictly positive definite functions of time. Equation (7) states that $d\nu_{\ell,t}$ can only take values 0 or 1. The conditional expectation $E(d\nu_{\ell,t}|\boldsymbol{\psi}_t, \boldsymbol{\psi}_t^\dagger)$ is called the compensator of the counting process $\nu_{\ell,t}$ and determines the jump rate given the values of the state vector and its complex adjoint at time $t$[42]. The equations governing $\boldsymbol{\psi}_t^\dagger$ follow immediately from (6) by applying the complex adjoint operation. We associate to (6) and to the equation for the adjoint, initial data on the Bloch hyper-sphere i.e., $\boldsymbol{\psi}_0^\dagger \boldsymbol{\psi}_0 = \|\boldsymbol{\psi}_0\|^2 = 1$. We emphasize that the stochastic Schrödinger equation (6) and the counting processes (7), (8) are essentially the same as in ref. [9].

Next, we need $\mu_t$ to obey an evolution law admitting solutions enjoying the martingale property. We thus require $\mu_t$ to evolve

according to the Itô stochastic differential equation

$$d\mu_t = \mu_t \sum_{\ell=1}^{\mathscr{L}}\left(\frac{\Gamma_{\ell,t}}{\imath_{\ell,t}} - 1\right) d\imath_{\ell,t} \tag{9}$$

$$d\imath_{\ell,t} = d\nu_{\ell,t} - \imath_{\ell,t}\left\|L_\ell \boldsymbol{\psi}_t\right\|^2 dt \tag{10}$$

$$\mu_0 = 1 \tag{11}$$

The solution of an equation of the form (9) is by construction a local martingale (see e.g., ref. [42] for details). Namely, the source of randomness in (9) are the innovation processes (10)[7,16] defined by compensating counting process increments $d\nu_{\ell,t}$'s by their conditional expectation (8). The immediate consequence is that the expectation value of the increments of $\mu_t$ conditional on the values of $\boldsymbol{\psi}_t$, $\boldsymbol{\psi}_t^\dagger$ vanishes at any time instant $t$:

$$E(d\mu_t|\boldsymbol{\psi}_t, \boldsymbol{\psi}_t^\dagger) = 0. \tag{12}$$

A local martingale becomes a strict martingale, i.e., satisfies the condition $E\mu_t = 1$ for all $t$, if the integrability condition $E \sup_t |\mu_t| < \infty$ holds. In practice, we expect $\mu_t$ to be a strict martingale if all $\Gamma_{\ell,t}$'s are bounded functions of $t$ during the evolution horizon. We take for granted this physically reasonable condition, and therefore that the process $\mu_t$ is a strict martingale.

The last step in the proof of the unraveling via the influence martingale is to compute the differential of the expectation value (5) using (6), (7), (8), and (9). A straightforward application of stochastic calculus proves that the state operator satisfies the time local master Eq. (1). We report the details of the calculation in Methods.

The question naturally arises whether the evolution law (6) preserves the squared norm of the stochastic process $\boldsymbol{\psi}_t$, and, as a consequence, justifies the interpretation of $\boldsymbol{\psi}_t$ as state vector of the system. We verify that the squared norm satisfies the Itô stochastic differential equation

$$d\left(\left\|\boldsymbol{\psi}_t\right\|^2\right) = \sum_{\ell=1}^{\mathscr{L}}\left(d\nu_{\ell,t} - \Gamma_{\ell,t}\left\|L_\ell \boldsymbol{\psi}_t\right\|^2 dt\right)\left(1 - \left\|\boldsymbol{\psi}_t\right\|^2\right). \tag{13}$$

Therefore, for arbitrary initial values $\boldsymbol{\psi}_0$, $\boldsymbol{\psi}_0^\dagger$ the expected value of the squared norm is not preserved unless (2) holds true. Nevertheless, the Bloch hyper-sphere (i.e., the manifold $\|\boldsymbol{\psi}_t\|^2 = 1$) is preserved by the dynamics. Thus, we can interpret $\boldsymbol{\psi}_t$ as a state vector for any quantum trajectory evolving from physically relevant initial data assigned on the Bloch hyper-sphere. A further useful consequence is that Bloch hyper-sphere valued solutions of (6) can always be couched into the form of the ratio of the solution of a linear stochastic differential equation divided by its norm. We defer the proof of the claim to Methods.

Some observations are in order regarding equations (6) and (9). The evolution law (9) of the influence martingale is enslaved to that of the state vector (6): $\mu_t$ exerts no feedback on the stochastic Schrödinger equation (6). Most importantly, the state vector is a Markov process and the influence martingale is also non anticipating. This is intuitively pleasing because in any finite dimensional Hilbert space the master equation (1) is just a matrix-valued time non-autonomous linear ordinary differential equation. Finally, we emphasize the different nature of the weights $\Gamma_{\ell,t}$'s, and of the positive definite rates $\imath_{\ell,t}$'s. The former ones are theoretical predictions fixed by the microscopic dynamics. The $\imath_{\ell,t}$'s are either inferred from experimental measurement or, in numerical applications, selected based on integration convenience. Such arbitrariness reflects the fact that quantum trajectories generated by an unraveling exist only contextually to a setup or in the language of[48] are "subjectively real".

Let us now turn to the interpretation of the influence martingale. Girsanov's formula (see e.g., ref. [42]) states that, when the martingale process $\mu_t$ in (5) is positive definite, it specifies a change of probability measure. The influence martingale can, however, take negative values when the $\Gamma_{\ell,t}$'s do so. As $\mu_t$ is non-anticipating, at any time $t$ it is always possible to represent it as the difference of two positive definite and correlated processes

$$\mu_t^{(\pm)} = \max(0, \pm\mu_t). \tag{14}$$

The immediate consequence is that we can couch (5) into the form

$$\boldsymbol{\rho}_t = \mathrm{E}\left(\mu_t^{(+)} \boldsymbol{\psi}_t \boldsymbol{\psi}_t^\dagger - \mu_t^{(-)} \boldsymbol{\psi}_t \boldsymbol{\psi}_t^\dagger\right) \tag{15}$$

Using the explicit expression of the influence martingale and of the state vector in terms of the fundamental solution of the linear dynamics ("Methods") it is straightforward to verify that the argument of the expectation value is the difference of two completely positive dynamical maps. We thus recognize that from the mathematical point of view, the need to introduce the influence martingale naturally stems from the general result in linear operator algebra known as the Wittstock–Paulsen decomposition[23] stating that any completely bounded map is always amenable to the difference of two completely positive maps.

If one insists on the change of probability measure interpretation, negative values of the influence martingale would imply that some realizations of quantum trajectories in the mathematical path-space should be weighed by a "negative probability". Exactly for the reasons put forward by Feynman in ref. [49], even such interpretation does not pose any logical difficulty when the initial state operator belongs to the compatibility domain of operators whose positivity is preserved by the evolution[33,34]. Negative values of the influence martingale only contribute as an intermediate step to the Monte Carlo evaluation of the state operator. In other words, they do not specify "the final probability of verifiable physical events"[49]. We refer to refs. [50,51] for a mathematically rigorous operational definition of negative probabilities in quantum mechanics recently developed starting from Feynman's argument.

From a more qualitative point of view, it is suggestive to interpret negative values of $\mu_t$ as a form of interference that occurs in the mathematical path-space in order to ensure the convergence of a Monte Carlo average to a completely bounded deterministic dynamical map. In ref. [49] (page 246–48) Feynman shows how interference patterns in a double slit experiment can be formally computed by means of arithmetic averages also including events weighed by a negative probability. The Lindblad weights $\Gamma_{\ell,t}$'s in the master equation are essentially time derivatives of survival probabilities. These latter quantities may take negative values in consequence of a phenomenon usually interpreted as re-scattering of decay products from the environment to the system[46]. Putting together these admittedly heuristic considerations, motivates the appeal of conceptualizing the influence martingale as the expression of a form of environment feedback-induced interference of quantum trajectories. By the same token, we motivate its name with a role reminiscent of the influence functional introduced by Feynman and Vernon in ref. [25]. To dispel any possible misunderstanding, however, we wish to add that the above considerations do not aim at arguing a one-to-one correspondence between the phenomenon of quantum interference and completely bounded maps.

**Examples.** In low dimensional Hilbert spaces, algorithms based on quantum trajectories are not expected to bring any numerical efficiency advantage with respect to direct integration of the

master equation. Thus the purpose of the examples is only to highlight how a non-anticipating unraveling reproduces physical phenomena such as quantum revivals usually attributed to memory and prescience effects. In all the examples we set for convenience $\imath_{\ell,t} = |\Gamma_{\ell,t}|$ in (8).

The master equation in Dirac's interaction picture of a two level atom in a photonic band gap[29,52] is

$$\dot{\boldsymbol{\rho}}_t = \frac{S_t}{2\imath}\left[\sigma_+\sigma_-, \boldsymbol{\rho}_t\right] + \Gamma_t\frac{\left[\sigma_-\boldsymbol{\rho}_t, \sigma_+\right] + \left[\sigma_-, \boldsymbol{\rho}_t\sigma_+\right]}{2} \tag{16}$$

where $\sigma_\pm = (\sigma_1 \pm \imath\,\sigma_2)/2$ and $\left\{\sigma_i\right\}_{i=1}^3$ are Pauli matrices. The time dependent functions $S_t$ and $\Gamma_t$ are respectively the Lamb shift and the Lindblad weight factor. Negative values of $\Gamma_t$ also imply a violation of the Kossakowski conditions (see e.g., refs. [22,47]) a weaker form of positivity that might be imposed on $\boldsymbol{\rho}_t$[53]. This fact renders the unraveling of (16) in quantum trajectories particularly probing.

In order to explore a genuine strong system-environment coupling, we proceed as in[38]. Let $\boldsymbol{g}$ and $\boldsymbol{e}$ be respectively the ground and excited state of $\sigma_+\sigma_-$. Using the solution of the off-diagonal matrix element $\boldsymbol{e}^\dagger\boldsymbol{\rho}_t\boldsymbol{g} = c_t\boldsymbol{e}^\dagger\boldsymbol{\rho}_0\boldsymbol{g}$ of a qubit in a photonic band gap (equation (2.21) of ref. [29] with $\beta = -\delta$), we determine the Lamb shift $S_t$ and weight factor $\Gamma_t$ by[2]

$$S_t = -2\,\mathrm{Im}\,\frac{\dot{c}_t}{c_t}, \;\; \Gamma_t = -2\,\mathrm{Re}\,\frac{\dot{c}_t}{c_t} \tag{17}$$

We also translate the origin of time to $t \approx 1.4$ so that $S_t$ and $\Gamma_t$ vanish at time origin.

In Fig. 1a, we show the time dependence of the Lindblad weight factor $\Gamma_t$ and the Lamb shift $S_t$. In Fig. 1c, d, we show typical realizations of $\mu_t$. In particular, Fig. 1d exhibits the exponential growth of $\mu_t$ when $\Gamma_t$ is negative and in the absence of jumps. On the other hand, when $\Gamma_t$ is positive $\mu_t$ is constant in between jumps. Finally, in Fig. 1c we show a realization of $\mu_t$ taking negative values in consequence of a quantum jump.

Figure 2 reports the result of our numerical integration for distinct values of the initial data. We generate the quantum

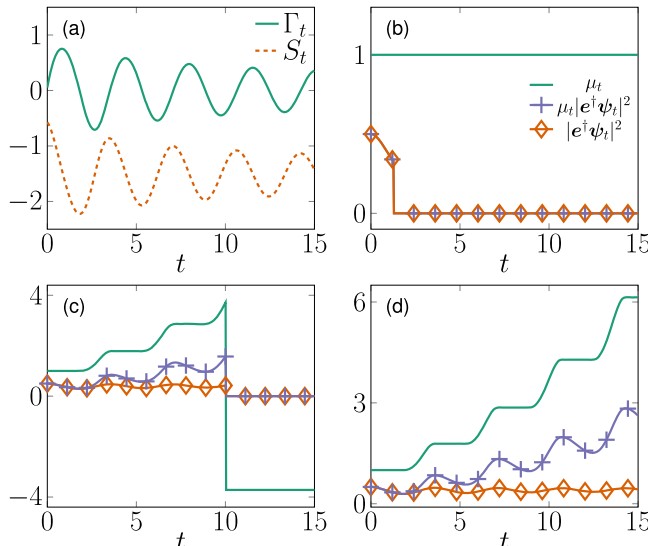

**Fig. 1 Illustration of qubit state and influence martingale trajectories.**
**a** Time dependence of the weight function $\Gamma_t$ (full line) and the Lamb shift $S_t$ (dashed), defined as in (17). **b–d** Different realizations of the stochastic evolution displaying $\left\|\boldsymbol{e}^\dagger\boldsymbol{\psi}_t\right\|^2$ (diamonds), $\mu_t\left\|\boldsymbol{\psi}_t\right\|^2$ (crosses), and $\mu_t$ (full line), where the system state $\boldsymbol{\psi}_t$ and influence martingale according to $\mu_t$ (6) and (9). The initial data are $\boldsymbol{\psi}_0 = (\boldsymbol{e} + \boldsymbol{g})/\sqrt{2}$ where $\boldsymbol{g}$ and $\boldsymbol{e}$ are respectively the ground and excited states of $\mathrm{H} = \sigma_+\sigma_-$.

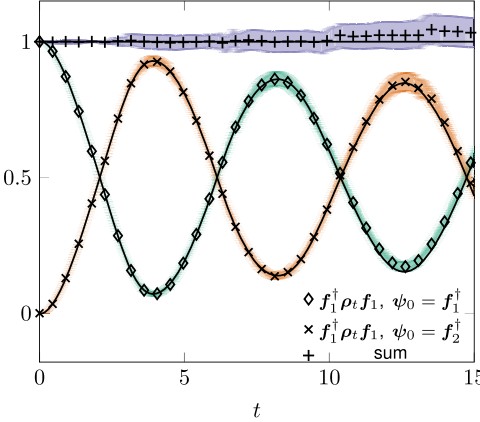

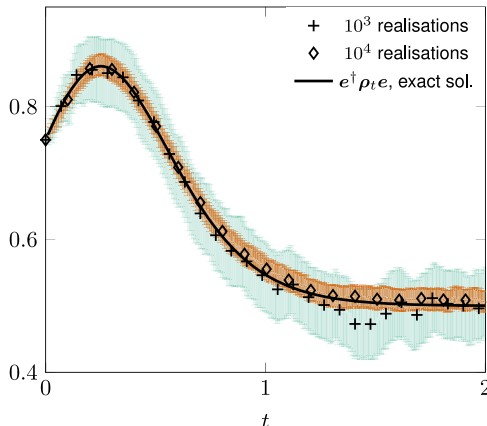

**Fig. 2 Illustration of the influence martingale for a qubit in a photonic band gap.** Monte Carlo averages versus master equation predictions for (16). The scatter plots show $\boldsymbol{f}_1^\dagger \boldsymbol{\rho}_t \boldsymbol{f}_1$, with $\boldsymbol{f}_1 = (\boldsymbol{e} + \boldsymbol{g})/\sqrt{2}$, with initial states $\boldsymbol{\psi}_0 = \boldsymbol{f}_1$ (diamonds) and $\boldsymbol{\psi}_0 = \boldsymbol{f}_2 = (\boldsymbol{e} - \boldsymbol{g})/\sqrt{2}$ (x's) and their sum (crosses), where $\boldsymbol{e}$ and $\boldsymbol{g}$ are the excited and ground state of $\sigma_+ \sigma_-$, respectively. The data is obtained from $10^4$ realizations. The continuous lines show the solutions obtained from directly integrating the master equation. The shaded area is determined by two times the square root of the ensemble variance of the indicator of interest. The starting time mesh for the adaptive code is $dt = 0.03$.

**Fig. 3 Illustration of the influence martingale for the random unitary model.** Monte Carlo averages versus master equation predictions for the energy level populations of the random unitary model (18). The Lindblad weights are $\Gamma_{1,t} = -0.5 + 2\tanh(\sqrt{2}\,t)$, $\Gamma_{2,t} = -1 + 2\tanh(\sqrt{3}\,t)$ and $\Gamma_{3,t} = -0.8 + 2\tanh(\sqrt{5}\,t)$. The initial condition of the qubit is $\boldsymbol{\psi}_0 = \frac{\sqrt{3}}{2}\boldsymbol{e} + \frac{1}{2}\boldsymbol{g}$ where $\boldsymbol{g}$, $\boldsymbol{e}$ are the ground and excited state of $\sigma_3$, respectively. The black full line gives the prediction by solving the master equation (18). The crosses show the stochastic average after $10^3$ realizations and the (light) green shaded area displays fluctuation-related errors estimated as in Fig. 2. Similarly, the diamonds show the ensemble average after $10^4$ realizations and the (dark) brown shaded area the estimated error.

trajectories by mapping (6) into a linear equation as described in Methods. The (black) full lines always denote predictions from the master equation. Monte Carlo averages are over ensembles of $10^4$ realizations. We theoretically estimate errors with twice the square root of the ensemble variance of the indicator of interest. In all cases, Monte Carlo averages and master equation predictions are well within fluctuation-induced errors. The occurrence of "quantum revivals" can be also quantitatively substantiated observing that the measure of system-environment information flow introduced in ref. [54] is simply related to $\Gamma_t$ for this model[47,54]. Namely, equation (66) of ref. [47] relates the direction of the information flow in the model to the sign of the time derivative of $|\Gamma_t|$. We perform the numerical integration using the Tsitouras 5/4 Runge-Kutta method automatically switching for stiffness detection to a 4th order A-stable Rosenbrock. The Julia code is offered ready for use in the "*DifferentialEquation.jl*" open source suite[55].

Next, we study a qubit model with "controllable positivity", whose dynamics are governed by the master equation

$$\dot{\boldsymbol{\rho}}_t = \sum_{\ell=1}^3 \Gamma_{\ell,t}(\sigma_\ell \boldsymbol{\rho}_t \sigma_\ell - \boldsymbol{\rho}_t). \qquad (18)$$

The above equation provides a mathematical model of an all-optical setup exhibiting controllable transitions from positive-divisible to non-positive divisible evolution laws[56]. We focus on the case when the Lindblad weights are of the form ($\ell = 1, 2, 3$)

$$\Gamma_{\ell,t} = -a_\ell + b_\ell \tanh(c_\ell t) \qquad (19)$$

We choose $a_\ell, b_\ell, c_\ell > 0$ such that all Lindblad weights are negative during a finite time interval around $t = 0$. As a consequence, not only the Kossakowski conditions are violated but the rate operator[57] has negative eigenvalues for the initial conditions we consider. Under these hypotheses, and to the best of our knowledge, only the influence martingale permits to unravel the master equation in quantum trajectories taking values in the Hilbert space of the system. Figure 3 shows the evolution in time of $\boldsymbol{e}^\dagger \boldsymbol{\rho}_t \boldsymbol{e}$ where $\boldsymbol{e}$ is the excited state of the Pauli matrix $\sigma_3$, i.e., it has eigenvalue 1. The full (black) line is the master equation prediction, the crosses show the Monte Carlo average (5) for $10^3$

realizations and the diamonds for $10^4$. The light green shaded area shows twice the square root of the variance for $10^3$ realizations and the darker brown for $10^4$. In agreement with the theory, increasing the number of realizations in the ensemble evinces convergence.

As a further example of how the influence martingale associates a quantum trajectory picture even to master equations with non positive definite solutions, we consider a Redfield equation model. We call Redfield a master equation of the form (1) obtained from an exact system-environment dynamics by implementing the Born–Markov approximation without a rotating wave approximation. Redfield equations are phenomenologically known to give accurate descriptions of the system dynamics at arbitrary system-environment coupling although they do not guarantee a positive time evolution of the state operator see e.g., refs. [2,30,58]. In "Methods", we outline the derivation of a Redfield equation from the exact dynamics of two non-interacting qubits in contact with a boson environment[59]. The result is an equation of the form (1) with only two Lindblad operators $\{L_\ell\}_{\ell=1}^2$ satisfying the commutation relations

$$[L_\ell, L_k^\dagger] = \delta_{\ell k}, \quad [L_\ell, L_k] = 0.$$

The corresponding weights are time independent

$$\Gamma_{\ell,t} = \lambda_\ell = \frac{\gamma_1 + \gamma_2 + (-1)^\ell \sqrt{2}\sqrt{\gamma_1^2 + \gamma_2^2 + 2\kappa^2}}{4} \qquad (20)$$

where $\gamma_1$, $\gamma_2$, $\kappa$ are real numbers. We refer to Methods for the explicit expressions of H and $\{L_\ell\}_{\ell=1}^2$. Inspection of (20) shows that $\lambda_1$ is negative definite. Figure 4 shows how the influence martingale reproduces the predictions of the Redfield equation.

**Numerical application.** Simulating Large Quantum Systems was the motivation of the authors of ref. [9] for unraveling a master equation or, in their words, for "Monte Carlo wave-function methods". The reason is the following. Given a $\mathcal{N}$-state system numerical integration of the master equation requires to store $O(\mathcal{N}^2)$ real numbers so that the computing time typically scales

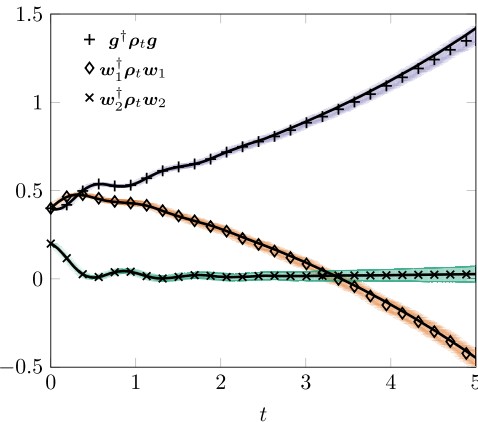

**Fig. 4 Illustration of the influence martingale for a Redfield equation.** The parameters for the model (20) are $\gamma_1 = 1$, $\gamma_2 = 4$, $\alpha = 3$ and $\kappa = 1$. The initial condition is $\psi_0 = \sqrt{0.2}\, w_1 + \sqrt{0.1}\, w_2 + \sqrt{0.7}\, g$ where $g$ is the ground state, $w_1 = L_1^\dagger g$ and $w_2 = L_2^\dagger g$. The crosses, diamonds, and x's show the result from Monte Carlo simulations after $10^4$ realizations. The full lines are master equation predictions and the shaded regions have the same meaning as in Fig. 2. The time mesh is $dt = 0.0125$. The diamonds show that $\rho_t$ is non positive definite for $t > 3$.

as $O(\mathcal{N}^4)$. Unraveling the state operator as the average over state vectors generated by a Markov process requires to store only $O(2\mathcal{N})$ real numbers for each realization. This means that the computing time scales as $O(\mathcal{N}^2 \times \mathcal{M})$ where $\mathcal{M}$ is the number of realizations or just $O(\mathcal{N}^2)$ on a parallel processor[3]. Thus, in high dimensional Hilbert spaces, also due to the existence of efficient numerical algorithms for stochastic differential equations[44], ensemble averages are expected to offer a real advantage for numerical computation[2,3,48]. It is worth noticing that the reasoning is very similar to that motivating the use of Lagrangian in place of Eulerian numerical methods in the context of classical hydrodynamics, see e.g., ref. [60].

To exhibit that the argument applies to the influence martingale, we compare computing times versus the dimension of the Hilbert space using QuTiP[43], a standard numerical toolbox for open quantum systems. Specifically, we take a chain of $\mathcal{N}$ coupled qubits and directly integrate the master equation using QuTiP. Next, we perform the same calculation using the QuTiP package for Monte Carlo wave-function methods combined with the implementation of the influence martingale. To do so we choose the rates in (8) as we detail below.

In (1) we take $\mathscr{L} = 2\mathcal{N}$ where the Lindblad operator $L_\ell$ for $\ell = 1, \dots, \mathcal{N}$ is the tensor product of the lowering operator of the $\ell$th qubit with the identity acting on the Hilbert space of the remaining qubits. Thus we set

$$L_\ell = \sigma_-^{(\ell)}, \quad L_{\ell+\mathcal{N}} = \sigma_+^{(\ell)} \quad \ell = 1, \dots, \mathcal{N} \tag{21}$$

The Hamiltonian is

$$H = \sum_{\ell=1}^{\mathcal{N}} \sigma_+^{(\ell)} \sigma_-^{(\ell)} + \lambda \sum_{\ell=1}^{\mathcal{N}-1} (\sigma_+^{(\ell)} \sigma_-^{(\ell+1)} + \sigma_+^{(\ell+1)} \sigma_-^{(\ell)}) \tag{22}$$

For the sake of simplicity, we assume that the Lindblad weights in (1) are all strictly positive definite with the only exception of $\Gamma_{1,t}$ which can take negative values. For any $\ell$ different from 1 and $\mathcal{N}+1$ we choose the rates of the counting processes (8)

$$\imath_{\ell,t} = \Gamma_{\ell,t} > 0 \quad \ell \neq 1, \mathcal{N}+1 \tag{23}$$

Next, we use the fact that given a collection of scalars $\omega_\ell$ on the real axis, it is always possible to find an equal number of positive definite scalars $\imath_\ell$ and a negative definite real $c$ such that the set of

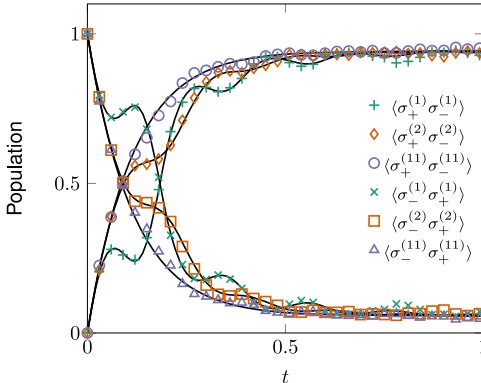

**Fig. 5 Site populations in a long qubit chain.** Population of sites 1, 4, and 11 for the qubit chain (22) with $L = 11$. The marks show the result of the stochastic evolution and the full black lines the result of numerically integrating the master equation. The weights are $\Gamma_{\ell,t} = \gamma$, $\Gamma_{\ell+N,t} = \delta$ for $\ell = 1, \dots N$, $\Gamma_{1+N,t} = \delta$ for all $t$, $\Gamma_{1,t} = \Gamma_{1,t} = \gamma - 12\exp(-2t^3)\sin^2(15t)$ with $\gamma = (1/0.129)(1 + 0.063)$ and $\delta = (1/0.129)0.063$ and $\lambda = 10$.

equations

$$\omega_\ell = \imath_\ell + c \tag{24}$$

is satisfied. We, therefore, set for any $t$

$$\Gamma_{1,t} = \imath_{1,t} + c_t \tag{25}$$

$$\Gamma_{1+\mathcal{N},t} = \imath_{1+\mathcal{N},t} + c_t \tag{26}$$

where now $\imath_{1,t}$ and $\imath_{1+\mathcal{N},t}$ specify the rates of the counting processes $d\nu_{1,t}$ and $d\nu_{1+\mathcal{N},t}$. As the Lindblad operators satisfy

$$\sigma_+^{(1)} \sigma_-^{(1)} + \sigma_-^{(1)} \sigma_+^{(1)} = 1_{\mathcal{H}} \tag{27}$$

on the Bloch hyper-sphere ($\|\psi_t\|^2 = 1$) we simplify the drift in (6) using the identity

$$\Gamma_{1,t} \frac{\sigma_+^{(1)} \sigma_-^{(1)} - \left\| \sigma_-^{(1)} \psi_t \right\|^2}{2} \psi_t + \Gamma_{1+\mathcal{N},t} \frac{\sigma_-^{(1)} \sigma_+^{(1)} - \left\| \sigma_+^{(1)} \psi_t \right\|^2}{2} \psi_t$$
$$= \imath_{1,t} \frac{\sigma_+^{(1)} \sigma_-^{(1)} - \left\| \sigma_-^{(1)} \psi_t \right\|^2}{2} \psi_t + \imath_{1+\mathcal{N},t} \frac{\sigma_-^{(1)} \sigma_+^{(1)} - \left\| \sigma_+^{(1)} \psi_t \right\|^2}{2} \psi_t \tag{28}$$

We conclude that the state operator evolves on the Bloch hyper-sphere according to the same stochastic Schrödinger equation of ref. [9]. We can therefore directly use the Monte Carlo wave function package of QuTiP for computing the evolution of the state vector. Using this information, we compute the influence martingale for each trajectory and finally the state operator. We refer to Methods for further details.

Figure 5 shows the populations of several sites for $\mathcal{N} = 11$. The computation time is shown in Fig. 6a. At $\mathcal{N} = 10$ we observe a cross-over of the computation time curves. After $\mathcal{N} = 11$ the influence martingale based algorithm becomes more efficient without applying any adapted optimization. From 12 qubits on, direct integration of the master equation becomes unwieldy (Apple M1 CPU). With the influence martingale, even 13 coupled qubits take only a few minutes of computation time. Most of the computation time is due to generating the trajectories. The actual computation of the martingale takes 0.19s for $\mathcal{N} = 2$ and 0.42s for $\mathcal{N} = 11$. The number of realizations is always $10^3$. Figure 6b shows the root mean square error of site occupations averaged over all sites. The error stays approximately constant when $\mathcal{N}$ increases.

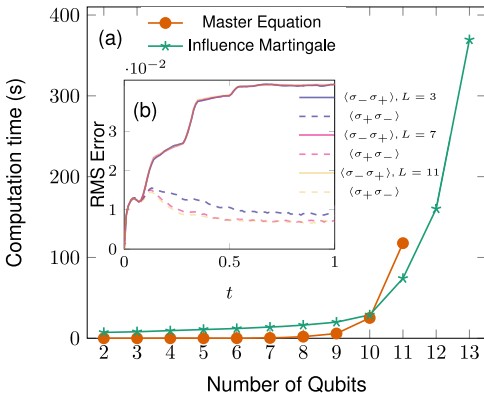

**Fig. 6 Numerical performance of the influence martingale. a** Computation time for both the Master Equation and Influence Martingale method as a function of the amount of qubits in the chain. For the stochastic method, we generated 1000 realizations. For $\mathcal{N} > 11$ the use of the master equation becomes unwieldy on our laptops whereas we were able to take further points using the unraveling. **b** The root mean square error of the populations averaged over all individual sites.

**Theoretical application**. In experimental quantum optics, photo-current is usually defined as the average number of detection events per unit of time. The stochastic differentials (7) mathematically describe increments of the photocurrent associated to detection events corresponding to transitions induced by the Lindblad operator $L_\ell$ in (1)[3].

The study of time-dependent transport properties of photo-excited undoped super-lattices highlighted the phenomenon of photo-current oscillations[61]. Photo-current oscillations naturally come about when working with the time convolutionless master equation with non positive Lindblad weights. In such a case and under standard hypotheses (see "Methods") it is straightforward to verify that a system governed by an Hamiltonian $H_o$ when isolated satisfies when the coupled to an environment the energy balance equation

$$d\operatorname{Tr}(H_o \boldsymbol{\rho}_t) = \frac{\operatorname{Tr}\big[H_o, H_t\big]\boldsymbol{\rho}_t dt}{\imath} + \sum_{\ell=1}^{\mathscr{L}} \epsilon_\ell \, d(E\mu_t \nu_{\ell,t}) \qquad (29)$$

where the $\epsilon_\ell$ are the energy quanta exchanged in transitions. The expression generalizes the results of ref. [3] showing that, as the strength of the system environment coupling increases, the average values of photo-current increments are modulated by the influence martingale in thermodynamic relations. The conclusion is that photo-current oscillations reflect a heat flow both from and to the system.

## Discussion

In this paper, we prove the existence of a non-anticipating unraveling of time local master equations whose fundamental solution is a completely bounded map. General results in operator algebra show that completely positive maps are a particular case of completely bounded ones. Accordingly, the unraveling we present naturally recovers the well known theory when the hypothesis of complete positivity is added. Specifically, the stochastic state vector obeys a Markovian evolution on the Bloch hyper-sphere of the system. The distinctive feature which makes the unraveling possible consists in realizing that outer products of the stochastic state vector must be weighed by a non-positive definite scalar martingale in order to generate Monte Carlo averages converging to a completely bounded map. This is a major conceptual difference with the martingale methods previously considered under the hypothesis of a completely positive divisible dynamics (see e.g., refs. [7,16,62]). There the martingale is

strictly positive because it is the norm squared of a complex vector. Furthermore, the introduction of a martingale weight factor serves the purpose of couching norm preserving state vectors obeying a non-linear evolution in terms of vectors satisfying linear stochastic differential equations.

In ref. [48] Wiseman discussed three interpretations of quantum trajectories generated by unravelings. Here, exactly as in[48], the order of listing is absolutely not meant to reflect importance. The first interpretation is as mathematical tool to compute the solution of the Lindblad–Gorini–Kossakowski–Sudarshan equation for high dimensional systems (see e.g., ref. [9]). The second interpretation of quantum trajectories construes them as subjectively real: their existence and features are determined only contextually to a given physical setup (see e.g., refs. [12,63]). And finally, quantum trajectories might be an element of a still missing theory of quantum state reduction[15,64–66]. We believe that the influence martingale yields a significant contribution to, at least, the first two interpretations. As a mathematical tool, the unraveling via influence martingale method only involves the use of ordinary stochastic differential equations driven by counting processes. Counting processes are paradigmatic mathematical models of measurement-induced state vector collapse as they naturally describe individual detection events while permitting straightforward numerical implementation[2,3,7,16]. The proof of the unraveling is rigorous and straightforward. The convergence of Monte Carlo averages is guaranteed by standard results in the theory of stochastic differential equations[44]. A further advantage is that the unraveling does not rely on any hypothesis on the sign of the scalar prefactors, weights, of the Lindblad operators in the master equation. Generalizing what was previously established for master equations derived in the weak coupling scaling limit[2,3], we here provide explicit evidence of the advantage of using the influence martingale to integrate a substantially larger class of master equations when the dimension of the Hilbert space is large. Finally, we observe that if the purpose of introducing quantum trajectories is limited to numerical applications, resorting to the generation of ostensible statistics as in ref. [67] may further speed up calculations. For ostensible statistics trace preservation in (5) holds not pathwise but only on average, Bloch hyper-sphere conservation is thus not required and the influence martingale can be replaced by a simple jump process.

Regarding the second interpretation, the meaning of the influence martingale is that of representing the completely bounded fundamental solution of the universal form of the time local master equation as an average over stochastic realizations of completely bounded maps. General results[23] in linear operator algebra prove that a completely bounded map can be embedded into a completely positive map acting on a larger Hilbert space. Combining this fact with the non-anticipating nature of the unraveling guarantees the measurement interpretation ("Methods"). An explicit example of the construction of the embedding is given in ref. [37]. There the unraveling was only defined in the extended Hilbert space thus requiring the introduction of a "minimal" dynamics for an auxiliary environment. Here we prove that quantum trajectories can be computed directly on the Bloch hyper-sphere of the system, their occurrence being always consistent with a measurement performed on an environment that does not need to be specified.

In conclusion, the main result of the present paper substantially extends the domain of application of quantum trajectory-based methods of state and dynamical parameter estimation, prediction and retrodiction as recently reviewed in ref. [68].

## Methods

**Derivation of the master equation**. In order to verify that the influence martingale representation of the state operator (5) generically yields a solution of the

master equation (1) we need to compute

$$d\boldsymbol{\rho}_t = d\,E\left(\mu_t \boldsymbol{\psi}_t \boldsymbol{\psi}_t^\dagger\right) \tag{30}$$

Differentiation commutes with the expectation value operation. Paths generated by stochastic differential equations have finite quadratic variation. We thus apply Itô lemma[42]

$$d(\mu_t \boldsymbol{\psi}_t \boldsymbol{\psi}_t^\dagger) = (d\mu_t)\boldsymbol{\psi}_t \boldsymbol{\psi}_t^\dagger + \mu_t d(\boldsymbol{\psi}_t \boldsymbol{\psi}_t^\dagger) + (d\mu_t) d(\boldsymbol{\psi}_t \boldsymbol{\psi}_t^\dagger) \tag{31}$$

and observe that the explicit evaluation of

$$d(\boldsymbol{\psi}_t \boldsymbol{\psi}_t^\dagger) = (d\boldsymbol{\psi}_t)\boldsymbol{\psi}_t^\dagger + \boldsymbol{\psi}_t(d\boldsymbol{\psi}_t^\dagger) + (d\boldsymbol{\psi}_t)(d\boldsymbol{\psi}_t^\dagger) \tag{32}$$

along the paths generated by (6) yields

$$\begin{aligned} d(\boldsymbol{\psi}_t \boldsymbol{\psi}_t^\dagger) = &- \imath[H_t, \boldsymbol{\psi}_t \boldsymbol{\psi}_t^\dagger]dt \\ &- \sum_{\ell=1}^{\mathscr{L}} \Gamma_{\ell,t}\left(\frac{L_\ell^\dagger L_\ell \boldsymbol{\psi}_t \boldsymbol{\psi}_t^\dagger + \boldsymbol{\psi}_t \boldsymbol{\psi}_t^\dagger L_\ell^\dagger L_\ell}{2} - \left\|L_\ell \boldsymbol{\psi}_t\right\|^2 \boldsymbol{\psi}_t \boldsymbol{\psi}_t^\dagger\right)dt \\ &+ \sum_{\ell=1}^{\mathscr{L}}\left(\frac{L_{\ell,t} \boldsymbol{\psi}_t \boldsymbol{\psi}_t^\dagger L_{\ell,t}^\dagger}{\left\|L_\ell \boldsymbol{\psi}_t\right\|^2} - \boldsymbol{\psi}_t \boldsymbol{\psi}_t^\dagger\right)d\nu_{\ell,t}. \end{aligned} \tag{33}$$

This last equation and the definitions of the counting process (7), (8) and influence martingale (9) differentials allow us to write

$$\begin{aligned} d(\mu_t \boldsymbol{\psi}_t \boldsymbol{\psi}_t^\dagger) = &(d\mu_t)\boldsymbol{\psi}_t \boldsymbol{\psi}_t^\dagger - \imath\mu_t[H_t, \boldsymbol{\psi}_t \boldsymbol{\psi}_t^\dagger]dt \\ &- \mu_t \sum_{\ell=1}^{\mathscr{L}} \Gamma_{\ell,t}\left(\frac{L_\ell^\dagger L_\ell \boldsymbol{\psi}_t \boldsymbol{\psi}_t^\dagger + \boldsymbol{\psi}_t \boldsymbol{\psi}_t^\dagger L_\ell^\dagger L_\ell}{2} - \left\|L_\ell \boldsymbol{\psi}_t\right\|^2 \boldsymbol{\psi}_t \boldsymbol{\psi}_t^\dagger\right)dt \\ &+ \mu_t \sum_{\ell=1}^{\mathscr{L}}\frac{\Gamma_{\ell,t}}{\imath_{\ell,t}}\left(\frac{L_{\ell,t} \boldsymbol{\psi}_t \boldsymbol{\psi}_t^\dagger L_\ell^\dagger}{\left\|L_\ell \boldsymbol{\psi}_t\right\|^2} - \boldsymbol{\psi}_t \boldsymbol{\psi}_t^\dagger\right)d\nu_{\ell,t}. \end{aligned} \tag{34}$$

Once we take the expectation value, the telescopic property of conditional expectations see e.g., ref. [42] and the martingale property (12) guarantee that all terms proportional to increments of the influence martingale vanish. The remaining terms under expectation reduce to

$$\begin{aligned} \dot{\boldsymbol{\rho}}_t = &- \imath[H_t, \boldsymbol{\rho}_t] \\ &- \sum_{\ell=1}^{\mathscr{L}} \Gamma_{\ell,t}\left(\frac{L_\ell^\dagger L_\ell \boldsymbol{\rho}_t + \boldsymbol{\rho}_t L_\ell^\dagger L_\ell}{2} - E\mu_t\left\|L_\ell \boldsymbol{\psi}_t\right\|^2 \boldsymbol{\psi}_t \boldsymbol{\psi}_t^\dagger\right) \\ &+ \sum_{\ell=1}^{\mathscr{L}} \Gamma_{\ell,t}E\mu_t\left(\frac{L_\ell \boldsymbol{\psi}_t \boldsymbol{\psi}_t^\dagger L_\ell^\dagger}{\left\|L_\ell \boldsymbol{\psi}_t\right\|^2} - \boldsymbol{\psi}_t \boldsymbol{\psi}_t^\dagger\right)\left\|L_\ell \boldsymbol{\psi}_t\right\|^2. \end{aligned} \tag{35}$$

Straightforward algebra then allows us to recover (1).

Finally, we emphasize that the proof relies on the stochastic nature of the quadratic variation of the martingale component of the counting process. Generically, the quadratic variation of a martingale is a stochastic process[42]. At variance with the general case, Lévy's characterization theorem proves that one of the defining properties of the Wiener process is self-averaging in mean square sense of the quadratic variation[42]. This fact hinders a straightforward extension of the above proof to quantum state diffusion[15]. Self-averaging of the quadratic variation is also a central element of the physics interpretation of quantum state diffusion. In fact, quantum state diffusion can be derived from the stochastic Schrödinger equation (6) in the singular limit of infinite number of detection events per unit of time[2] and is, in this sense, adapted to describe a more restrictive class of physics contexts.

**Linear stochastic differential equation.** The Itô stochastic differential equation (6) preserves the Bloch hyper-sphere. On the hyper-sphere, we can look for solutions of the stochastic Schrödinger equation (6) of the form

$$\boldsymbol{\psi}_t = \frac{\boldsymbol{\varphi}_t}{\left\|\boldsymbol{\varphi}_t\right\|} \tag{36}$$

The change of variables (36) maps (6) into the linear problem

$$\begin{aligned} d\boldsymbol{\varphi}_t = &- \imath\,H_t \boldsymbol{\varphi}_t\,dt \\ &- \sum_{\ell=1}^{\mathscr{L}}\left(\frac{\Gamma_{\ell,t}}{2} L_\ell^\dagger L_\ell dt - d\nu_{\ell,t}(L_\ell - 1)\right)\boldsymbol{\varphi}_t \end{aligned} \tag{37}$$

Once we know $\boldsymbol{\varphi}_t$, we can use (36) to determine the state vector and the influence martingale. In particular, the influence martingale always admits the factorization

$$\mu_t = \exp\left(\int_0^t ds\sum_{\ell=1}^{\mathscr{L}}\left(\imath_{\ell,s} - \Gamma_{\ell,s}\right)\left\|L_\ell \boldsymbol{\psi}_s\right\|^2\right)\bar{\mu}_t \tag{38}$$

where $\bar{\mu}_t$ is a pure jump process. This factorization is of use in numerical implementations. From (38) we readily see that negative values of the $\Gamma_{\ell,t}$'s exponentially enhance the contribution of the realization of the state vector to the expectation value.

## Derivation of the stochastic Wittstock–Paulsen decomposition

*Explicit expression of the probability measure.* We describe sequences of detection events by first supposing that a fixed number $n$ of jumps occurs in the time interval $(t_o, t]$. Next, we suppose that a jump of type $\ell_i$ with $\ell_i$ taking values in $\{1, 2, \dots, \mathscr{L}\}$ occurs at time at $s_i$ satisfying for $i = 1, \dots, n$ the chain of inequalities

$$t > s_n > \dots > s_i > \dots > s_1 > s_o = t_o \tag{39}$$

An arbitrary sequence of detection events $\omega$ thus corresponds to a $2n$-tuple $\{\ell_i, s_i\}_{i=1}^n$. The total number $n$ of jumps ranges from zero to infinity. We refer to this mathematical description of events as "waiting time representation". On the Bloch hyper-sphere, all information about the dynamics of (6) in a time interval $[s, t]$ during which no jump occurs is encapsulated in the Green function of the linear dynamics (37)

$$\dot{G}_{ts} = -\imath\,H_t G_{ts} - \frac{1}{2}\sum_{\ell=1}^{\mathscr{L}} \Gamma_{\ell,t} L_\ell^\dagger L_\ell G_{ts} + \delta(t-s) \tag{40}$$

$$\lim_{t\searrow s} G_{ts} = 1_{\mathcal{H}} \tag{41}$$

Exactly repeating the same steps as in § 6.1 of ref. [2], we obtain the expression of the state vector conditional upon $\omega$

$$E\left(\boldsymbol{\psi}_t|\omega = \{\ell_i, s_i\}_{i=1}^n, \boldsymbol{\psi}_{t_o} = z\right) = \frac{\Lambda_{tt_o}\left(\{\ell_i, s_i\}_{i=1}^n\right)z}{\left\|\Lambda_{tt_o}\left(\{\ell_i, s_i\}_{i=1}^n\right)z\right\|}. \tag{42}$$

By using the identity

$$e^{-\int_s^t du\sum_{\ell=1}^{\mathscr{L}} \imath_{\ell,u}\frac{\|L_\ell G_{us}z\|^2}{\|G_{us}z\|^2}} = \left\|G_{ts}z\right\|^2 e^{-\int_s^t du\sum_{\ell=1}^{\mathscr{L}}(\imath_{\ell,u}-\Gamma_{\ell,u})\frac{\|L_\ell G_{us}z\|^2}{\|G_{us}z\|^2}} \tag{43}$$

we find that the multi-time probability density of the conditioning event is equal to

$$p_{tt_o}(\omega = \{\ell_i, s_i\}_{i=1}^n|\boldsymbol{\psi}_{t_o} = z) = \delta_{n,0}\frac{\left\|G_{tt_o}z\right\|^2}{m_{tt_o}(z)} + \frac{(1-\delta_{n,0})\left\|\Lambda_{tt_o}\left(\{\ell_i, s_i\}_{i=1}^n\right)z\right\|^2}{m_{ts_n}\left(\prod_{j=1}^n L_{\ell_j} G_{s_js_{j-1}}z\right)\cdots\,m_{s_1t_o}(z)} \tag{44}$$

In writing (42), (44) we introduced the tensor valued process

$$\Lambda_{tt_o}\left(\{\ell_i, s_i\}_{i=1}^n\right) := \delta_{n,0}G_{tt_o} + (1-\delta_{n,0})G_{ts_n}\prod_{i=1}^n \sqrt{\imath_{\ell_i,s_i}}L_{\ell_i} G_{s_is_{i-1}} \tag{45}$$

and the scalar

$$m_{ts}(z) := e^{\int_s^t du\sum_{\ell=1}^{\mathscr{L}}(\imath_{\ell,u}-\Gamma_{\ell,u})\frac{\|L_\ell G_{us}z\|^2}{\|G_{us}z\|^2}} \tag{46}$$

which is the value of the influence martingale if no jumps occur in the interval $[s, t]$. In order to neaten the notation in (42) and (44) and below, we omit to write the condition $\boldsymbol{\psi}_{t_o}^\dagger = z^\dagger$ that fully specifies the initial data on the Bloch hyper-sphere.

If we now restrict the attention to the computation of quantum probabilities, we see that the product of the pure state operator specified by (42) times its probability (44) yields the stochastic dynamical map

$$\Phi_{tt_o}[\omega = \{\ell_i, s_i\}_{i=1}^n](zz^\dagger) = \frac{\Lambda_{tt_o}\left(\{\ell_i, s_i\}_{i=1}^n\right)zz^\dagger\Lambda_{tt_o}^\dagger\left(\{\ell_i, s_i\}_{i=1}^n\right)}{m_{ts_n}\left(\prod_{j=1}^n L_{\ell_j} G_{s_js_{j-1}}z\right)\cdots\,m_{s_1t_o}(z)} \tag{47}$$

satisfying by construction the unit trace condition

$$\int \mathfrak{m}(d\omega)\text{Tr}\Phi_{tt_o}[\omega](zz^\dagger) := \sum_{n=0}^\infty \prod_{i=1}^n \sum_{\ell_i=1}^{\mathscr{L}} \int_{t_o}^t ds_i p_{tt_o}(\omega = \{\ell_i, s_i\}_{i=1}^n) = 1. \tag{48}$$

Thus the dynamical map (47) takes the form of the generalized operator sum representation[22,69]. It differs from the Choi representation of a completely positive map (see e.g., ref. [23]) in consequence of the non-linear dependence upon the initial state.

*Stochastic Wittstock–Paulsen decomposition.* Multiplying (47) by the influence martingale (38) occasions the cancellation of any non-linear dependence upon the initial data

$$E(\mu_t \boldsymbol{\psi}_t \boldsymbol{\psi}_t^\dagger|\omega, \boldsymbol{\psi}_{t_o} = z) = \Lambda_{tt_o}^{(+)}(\omega)zz^\dagger\Lambda_{tt_o}^{(+)\dagger}(\omega) - \Lambda_{tt_o}^{(-)}(\omega)zz^\dagger\Lambda_{tt_o}^{(-)\dagger}(\omega) \tag{49}$$

with

$$\Lambda_{tt_o}^{(\pm)}\left(\{\ell_i, s_i\}_{i=1}^n\right) = \left(\frac{\max(0, \pm\prod_{k=1}^n \Gamma_{\ell_k,s_k})}{\prod_{j=1}^n \imath_{\ell_j,s_j}}\right)^{1/2}\Lambda_{tt_o}\left(\{\ell_i, s_i\}_{i=1}^n\right) \tag{50}$$

We thus verify that (15) is indeed an expectation value over the difference of completely positive stochastic dynamical maps.

**Derivation of the Redfield equation for a two-qubit system**. We consider two non-interacting qubits in contact with an environment consisting of $\mathcal{N} \uparrow \infty$ bosonic oscillators

$$\hat{H} = \sum_{i=1}^{2} \omega^{(i)} \sigma_+^{(i)} \sigma_-^{(i)} + \sum_{k=1}^{\mathcal{N}} \left( \epsilon_k b_k^\dagger b_k + \sum_{i=1}^{2} g_k \left( \sigma_+^{(i)} b_k + b_k^\dagger \sigma_-^{(i)} \right) \right)$$

Here $\sigma_\pm^{(i)}$ is the lift to the tensor product Hilbert space of the ladder operators acting on the Hilbert space of individual qubits $i = 1, 2$. Tracing out the environment yields the master equation[59]

$$\dot{\boldsymbol{\rho}}_t = -\imath \sum_{i,j=1}^{2} A_{ij} [\sigma_+^{(j)} \sigma_-^{(i)}, \boldsymbol{\rho}_t] + \sum_{i,j=1}^{2} B_{ij} \frac{\left[ \sigma_-^{(i)}, \boldsymbol{\rho}_t \sigma_+^{(j)} \right] + \left[ \sigma_-^{(i)} \boldsymbol{\rho}_t, \sigma_+^{(j)} \right]}{2} \quad (51)$$

where the $A_{ij}$'s and $B_{ij}$'s are respectively the components of the matrix

$$A = \begin{bmatrix} \alpha & \alpha + \frac{\kappa}{2} - \imath \frac{\gamma_2 - \gamma_1}{4} \\ \alpha + \frac{\kappa}{2} - \imath \frac{\gamma_1 - \gamma_2}{4} & \alpha + \kappa \end{bmatrix} \quad (52)$$

and

$$B = \frac{1}{2} \begin{bmatrix} \gamma_1 & \frac{\gamma_1 + \gamma_2}{2} - \imath \kappa \\ \frac{\gamma_1 + \gamma_2}{2} + \imath \kappa & \gamma_2 \end{bmatrix} \quad (53)$$

The key observation is that the numbers $\gamma_1, \gamma_2$ are positive and $\alpha, \kappa$ are real. Thus, the matrix B is self-adjoint, and it is therefore unitarily equivalent to a real diagonal matrix

$$\text{diag } B = U^\dagger B U = \begin{bmatrix} \lambda_1 & 0 \\ 0 & \lambda_2 \end{bmatrix} \quad (54)$$

where for $i = 1, 2$

$$\lambda_i = \frac{\gamma_1 + \gamma_2}{4} + (-1)^i \sqrt{\frac{\gamma_1^2 + \gamma_2^2 + 2\kappa^2}{8}} \quad (55)$$

Upon defining the Lindblad operators

$$L_j = \sum_{i=1}^{2} \sigma_-^{(i)} U_{ij}, \quad j = 1, 2 \quad (56)$$

and the self-adjoint operator

$$H = \sum_{i,j=1}^{2} A_{ij} \sigma_+^{(j)} \sigma_-^{(i)} \quad (57)$$

we finally arrive at the master equation

$$\dot{\boldsymbol{\rho}}_t = -\imath [H, \boldsymbol{\rho}_t] + \sum_{\ell=1}^{2} \lambda_\ell \frac{\left[ L_\ell, \boldsymbol{\rho}_t L_\ell^\dagger \right] + \left[ L_\ell \boldsymbol{\rho}_t, L_\ell^\dagger \right]}{2} \quad (58)$$

**QuTiP implementation**. For the numerics, we assign the Lindblad weights to be

$$\Gamma_{1,t} = \gamma - 12 \exp(-2t^3) \sin^2(15t) \quad (59)$$

and

$$\Gamma_{\ell,t} = \gamma > 0 \quad \ell = 2, \dots, \mathcal{N} \quad (60)$$

$$\Gamma_{\ell+\mathcal{N},t} = \delta > 0 \quad \ell = 1, \dots, \mathcal{N} \quad (61)$$

In the time interval $[0.2, 0.25]$ we choose a solution of the over-determined system

$$\Gamma_{1,t} = \imath_{1,t} + c_t \quad (62)$$

$$\delta = \imath_{1+\mathcal{N},t} + c_t \quad (63)$$

by defining

$$c_t = -\frac{(1 - \text{sign}(\Gamma_{1,t}))}{2} \frac{\Gamma_{1,t} - \delta/2}{\Gamma_{1,t} - \delta} \imath_{1+\mathcal{N},t}. \quad (64)$$

The choice of $c_t$ is merely based on the empirical observation that for the values of $\gamma$ and $\delta$ we used, the resulting values of $\imath_{1+\mathcal{N}}, \imath_1$ are positive and efficiently handled by QuTiP.

**Photo-current oscillations**. One way to conceptualize time-convolutionless perturbation theory is as an avenue to implement at any order in the system environment coupling constant a Markov approximation in the derivation of the master equation. The leading order corresponds to the weak coupling approximation. In that case, the Lindblad operators $\{L_\ell\}_{\ell=1}^{\mathcal{L}}$ are obtained as eigenoperators of the unperturbed isolated system Hamiltonian $H_o$[2]:

$$[H_o, L_\ell] = \epsilon_\ell L_\ell \quad (65)$$

for some $\epsilon_i$'s, taking positive and negative values. We straightforwardly verify that

$$\left[ H_o, L_\ell^\dagger L_\ell \right] = 0 \quad (66)$$

immediately follows. In general, $H_o$ differs from the Hamilton operator $H_t$ in (1) by Lamb shift corrections. It is then reasonable to surmise that higher order corrections determined by time convolutionless perturbation theory only affect the intensity of the Lamb shift and the values of weight functions $\Gamma_{\ell,t}$ of the Lindblad operators in (1) see e.g., ref. [70]. Under these hypotheses a straightforward calculation using

$$dE(\mu_t \nu_{\ell,t}) = \Gamma_{\ell,t} \text{Tr} L_\ell \boldsymbol{\rho}_t L_\ell^\dagger dt \quad (67)$$

allows us to derive the energy balance equation (29).

**Measurement interpretation**. The mathematical notion of "instrument" $\mathcal{I}$ provides the description of quantum measurement adapted to quantum trajectory theory see e.g., ref. [7]. An instrument is a map from a classical probability space $(\Omega, \mathcal{F}, dP)$[42] to the space of bounded operators acting on a Hilbert space $\mathcal{H}$ that for any pre-measurement state operator $X$ satisfies

$$\mathcal{I}_t(F)[X] = \sum_k \int_F dP(\omega) \mathcal{V}_{t,k}(\omega) X \mathcal{V}_{t,k}^\dagger(\omega) \quad (68a)$$

$$\sum_k \int_F dP(\omega) \mathcal{V}_{t,k}^\dagger(\omega) \mathcal{V}_{t,k}(\omega) = 1_{\mathcal{H}} \quad (68b)$$

Here $dP$ is a classical probability measure, $F \subseteq \Omega$ is an event in the $\sigma$-algebra $\mathcal{F}$ describing all possible outcomes from the sample space $\Omega$ and $\mathcal{V}_{t,k}(\omega)$ are operators acting on $\mathcal{H}$. In (68a) we regard the instrument as a function of the time $t$.

In order to make contact with quantum trajectories unraveling a completely bounded state operator, we interpret $\mathcal{H}$ as an embedding Hilbert space $\mathcal{H} = \mathcal{H}_E \otimes \mathcal{H}_S$ and $X = \pi(\boldsymbol{\rho}_0)$ as a representation of the initial state operator of the system onto $\mathcal{H}$. Finally let $\left\{ E_{ij} \right\}_{i,j=1}^{\dim \mathcal{H}_E}$ be the canonical basis of the space of operators acting on $\mathcal{H}_E$. We assume $\dim \mathcal{H}_E < \infty$ and recall that $E_{ij} = e_i e_j^\dagger$ where $\left\{ e_i \right\}_{i=1}^{\dim \mathcal{H}_E}$ is the canonical basis of $\mathcal{H}_E$ itself. Let now O be a self-adjoint operator acting on $\mathcal{H}_S$. General results in linear operator algebra[23] ensure the existence of an instrument such that for $i \neq j$ we can write

$$\text{Tr}_{\mathcal{H}_S} O\left( \Lambda_{t t_o}^{(+)}(\omega) \boldsymbol{\rho}_0 \Lambda_{t t_o}^{(+)\dagger}(\omega) - \Lambda_{t t_o}^{(-)}(\omega) \boldsymbol{\rho}_0 \Lambda_{t t_o}^{(-)\dagger}(\omega) \right)$$

$$= \sum_k \text{Tr}_{\mathcal{H}} \left( \frac{E_{ij} + E_{ij}^\dagger}{2} \otimes O \mathcal{V}_{t t_o, k}(\omega) X \mathcal{V}_{t t_o, k}^\dagger(\omega) \right) \quad (69)$$

for $\Lambda_{t t_o}^{(\pm)}$ defined in (50) and some $\mathcal{V}_{t t_o, k}(\omega)$. We refer to ref. [37] for an explicit construction of an embedding representation.

## Data availability

No datasets were generated or analyzed during the current study.

## Code availability

The code used to generate Figs. 5 and 6 is available on https://github.com/QuBrecht/Influence-Martingale. The code can be used to implement any of the other examples. Code Available Upon Request.

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

## Acknowledgements

We warmly thank Alberto Barchielli, Joachim Ankerhold, Jukka Pekola, and Dmitry Golubev for useful comments and discussions. B.D. acknowledges financial support from the AtMath collaboration at the University of Helsinki. The authors would like to dedicate this paper to the memory of Krzysztof Gawędzki from whom they learned how to conceptualize martingales in statistical physics.

## Author contributions

P.M.-G. conceived the original idea. P.M.-G. and B.D. equally contributed to the further refinement and development of the results.

## Funding

## Competing interests

The authors declare no competing interests.
