## [Peer Review File · Nature Communications]

Quantum trajectory framework for general time-local master equationsREVIEWER COMMENTS

Reviewer #1 (Remarks to the Author):

Donvil and Muratore-Ginanneschi present a new way of simulating quantum trajectories of open quantum systems. Specifically, they look at non-Markovian open system, non-Markovian in the sense that they are described by rates in the Lindblad master equation that are not positive for all times.

Their new method involves the introduction of an “influence martingale”, slave to the state vector. The evolution of this martingale is key to then predicting the evolution of the averaged density operator.

Though an interesting new perspective, I do not believe this technique brings a method that enables calculations that could not have been done previously. Indeed, there are already many methods for simulating non-Markovian dynamics, including a quantum jumps method (ref 29 of the current paper). In addition I do not think that this new technique brings any obvious new insight into this kind of quantum dynamics. Though I think the paper should be published somewhere, I cannot therefore recommend publication in Nature Communications.

In general, the authors should consider:

- * What calculation does this open that could not have been done with other methods?
- * How does the computation time for this method compare to other techniques?
- * Are there any new insights that this method brings?

I also have some specific comments:

1. In Eq. 1, I would state specifically that the Gamma can be time-dependent.
2. In the discussion of P3, first column - this is reliant on looking at the other papers referenced - it is not possible to follow what the authors have done, even as an expert, without reference to these other papers. I therefore think the authors need to put in more detail here.
3. In Figs. 2 and 4 - though the agreement does seem to be good between the new method and the master equation there does, nonetheless, seem to be some systematic disagreement. Have the authors checked that the results will eventually converge? If they are not exactly converging - why is this?
4. P3, col 2: the authors discuss information back flow - can they plot a measure of this, so that it can be verified for this case?
5. P4. In the last sentence of the conclusion, the authors say that the influence martingale yields a significant contribution to the first two of their three interpretations. Can the authors expand on this, especially on how it does this for their second interpretation?

Reviewer #2 (Remarks to the Author):

The manuscript introduces a novel quantum-jump description of the general dynamics of open quantum systems. The basic idea is to weigh the pure states generated by the stochastic differential equation [Eq.(3) of the manuscript] with a local martingale, whose evolution is in turn fixed by the evolution of the state vector [see Eq.(5a)]. The key point is that such a martingale can take negative

values, when the coefficients of the master equation do so, which allows one to get the correct average for general dynamics, i.e., including non-Markovian ones where at least one of the master equation coefficients is negative and thus the original Monte Carlo Wave Function description does not apply. Compared to other non-Markovian quantum-jump descriptions present in the literature, the strong advantage of the method introduced here is that the negativity of the martingale accounts for all the non-Markovianity, while the stochastic differential equation for the state vector is the “standard” one, fixed by independent Poisson processes whose conditional expectation values depend on the absolute value of the master equation, as to ensure their positivity. The examples discussed in the second part of the manuscript explicitly show the applicability of the method to relevant physical systems, further accounting for a possible difficulty associated with the stiffness of the stochastic dynamics.

I think that the quantum-jump method introduced in the manuscript is extremely interesting and potentially very powerful. The use of a properly defined martingale, whose evolution is determined by the stochastic evolution of the state vector, represents in fact a significant novelty within the context of the quantum-jump descriptions of general open quantum system dynamics. Besides its mathematical and numerical relevance, such an approach has also quite an interesting physical interpretation, where the role of the martingale is to determine an interference among the different trajectories, analogously to what happens with the Feynman-Vernon influence functional. Quite interestingly, the average dynamics can then be seen as due to an interference among different stochastic trajectories and not simply as a probabilistic mixture of them, as in standard quantum-jump approaches. Last but not least, the manuscript is very clearly written, and its conclusions are strongly motivated both from the mathematical and the physical point of view.

In conclusion, I think that the manuscript provides a certainly valuable contribution to the theory of open quantum systems, and will then have a significant impact not only in this field, but also in the many areas of research relying on the realistic description of the dynamics of quantum systems. I then recommend the paper for publication in Nature Communications, and I only have the following (minor) remarks.

- If I am not missing something, the quantum trajectories defined by Eqs.(3)-(5) would be able to reproduce an evolution of the average state as fixed an equation (1) even if such an evolution is not completely positive (and actually not even positive); this is essentially due to the fact that the average in Eq.(6) is made with a weight – the martingale μ_t – that can be negative. To my opinion, this is not a limitation of the method – since in quantum unravelings the starting point is the master equation itself -- but rather possibly an advantage; anyway, if I am correct, I would suggest the authors to mention this point, especially in view of the comparison with other quantum-jump methods.

- In a way related with the previous point, before Eq.(2), instead of saying that the Lindblad master equation follows from the postulate of quantum mechanics if and only if Eq.(2) hold, I would rather mention explicitly the requirement of complete positivity of the dynamics as well as of its propagators (one might even directly mention the property of CP-divisibility, which is quite well-established in the literature).

- In the second column of page 1, 10 lines from the end, please note that “introduction an extra” should read “introduction of an extra”.

- In the caption of Fig.1, shouldn't μ_t be referred simply to the solid line, rather than to the x-shapes?

- It might be convenient to include the description of the symbols also in the caption of Fig.2.

Reviewer #3 (Remarks to the Author):

Let me start by saying that while I work on quantum stochastic processes, I do not work on quantum trajectories.

The paper is concerned with the breakdown of quantum trajectories framework for simulating complex quantum dynamics in presence of non-Markovianity. In particular, when divisibility of the dynamics breaks down the GKSL master equation does not preserve positivity and leads to interpretation issues.

The gist of the present manuscript is to introduce a slave master equation for a weight that ensures the density of state is well-behaved in time. The slave master equation is shown to have positive and negative elements, which the authors interpret as interference.

There are several elements of the paper that are not clear to me. I will state list my objections below.

1. I do not see how Eq. 5 is justified. It's not derived in the Methods section.
2. I am not sure what is happening in the Methods section as it relies on Eqs 3,4,5.
3. I am not convinced of the "interference" interpretation of the slave master equation. Afterall, it's just adjusting weights. Often quantum states must be decomposed with affine weights, but it may not mean interference. Can they provide more physical argument? I am also skeptical since quantum trajectories themselves are not physically real in a sense....
4. The Numerics section of the paper was really difficult to penetrate. I am sure the numerics are all fine, but what should I get out of this? The authors don't seem to compare to previous methods, which presumably fail. How does the method compare to exact calculations? It's not clear how difficult are problems that they do consider for numerics.

Overall, I think the paper is not easy to read, even for an expert in quantum stochastic processes. I think the contributions may be significant, but I was unable to ascertain this from the paper itself.

we hereby resubmit an extensively revised version of our manuscript "Interference of Quantum trajectories". The revision, though extensive, does not concern the content of the results and in particular of the main result of the paper, essentially a theorem, as none of the reviewers questioned their being correct, relevant and novel. The extensive revision focuses on the presentation of the results and emphasizes the potential for new applications that our results open. Furthermore, we discuss two new examples that, to the best of our knowledge, no other quantum trajectory picture can handle in the Hilbert space of the system. Finally, we now make explicit reference to an essay of R.P. Feynman which corroborates and greatly facilitates the physical interpretation of our results. We also substantially expanded Methods to offer a step-by-step guidance to the proof of the main result. The proof relies only on a straightforward application of stochastic calculus for Poisson noise. We also briefly recall the content of Itô lemma (i.e. stochastic differentiation). Following the instructions, we marked in red changes in the text. A detailed account of the changes in the paper is contained in the response to the reviewers.

Reply to Reviewer #1

We thank the reviewer for carefully going through our paper. We are very pleased that the reviewer acknowledges that our results are new and correct. We are however somewhat surprised by the reviewer recommendation. In our view the recommendation overlooks the fact that our contribution represents a change of paradigm (unraveling by ordinary Markov processes instead than unraveling by processes with memory) rather than an incremental contribution with respect to the existing literature and in particular ref [30] of the revised version (Piilo et Al PRL 2008. This was ref [29] in the old version, below we refer to articles according to the numbering in the amended version). We have improved the presentation to emphasise the novelty of our method and how it substantially differs from earlier works [28] and [30].

To start with, the reviewer asks the following general questions.

1. What calculation does this open that could not have been done with other methods?
 - Our method shows that the state vector unraveling a time local master equation always obeys an ordinary stochastic differential equation with jumps taking values in the Hilbert space of the closed system. Consequently, the state vector is always a Markov process. We emphasize here that our method encompasses quantum maps which are just divisible and therefore not preserving the positivity of the state operator. Positivity plays no role in our derivation. As a concrete illustration, we now include in examples 3 and 4 cases that cannot be treated even using the rate operator formalism of ref [32] (Smirne et al. PRL 2020) using the ideas of ref [30]. The fact that the influence martingale relies on simple explicit equations makes the method highly flexible. It can therefore encompass very general physical models. For example, as we outline in the discussion, in a forthcoming work we show how the influence martingale enables us to unravel even hybrid master equations depending on classical stochastic indicators associated to collective variables associated to the

state of the environment.

We give here some further details. We can use the state vector equation together with the influence martingale equation and a stochastic differential equation for the energy of the environment to unravel the exact master equation governing the evolution of the state operator of a central fermion defined by tracing out finite sized environment in the presence of an external source coupled to the energy operator of the environment. The unraveling holds for arbitrary sized environments. Computing the explicit form of the jump rates in the energy process requires inverting Laplace transforms which can be done in a fully analytic way in specific cases. Doing this calculation was the original motivation to introduce the influence martingale. We are not aware of any way how this calculation could be done by any other method. The reason is that solution of a stochastic differential equation is a Markov process so it is straightforward to establish a correspondence e.g. between jumps in the state vectors and energy exchanges with and within the reservoir. The fact that quantum trajectories are Markov processes has also subtle consequences for the interpretation on which we return below.

2. How does the computation time for this method compare to other techniques?

- The influence martingale method purely relies on the use of ordinary stochastic differential equations with jumps. The proof of the method is rigorous and elementary inasmuch it follows from a straightforward application of Itô lemma. These facts guarantee not only the convergence of Monte Carlo averages as a proven mathematical result but also permit to directly apply a well developed theory for efficient algorithmic implementations. In the amended version we refer to the monograph Eckhard Platen and Nicola Bruti-Liberati *Numerical Solution of Stochastic Differential Equations with Jumps in Finance* Springer 2010 (ref [47]).

The unraveling proposed by Breuer, Kappler, and Petruccione, ref [29] is also purely based on ordinary stochastic differential equations with jumps. The major difference is that [29] requires the introduction of a ghost Hilbert space of the same dimension of the physical Hilbert space of the system. The duplication of degrees of freedom, evolution equations etc. unavoidably leads to longer computation times and larger memory requirements.

Coming finally to [30], the general functional dependence among realizations i.e. the equation satisfied by the stochastic state vector is not explicitly given. What is shown in [30], see also [32] is that in all the considered examples there exists an algorithm which unravels the master equation by keeping track of events occurred during distinct realizations of a stochastic process. In the absence of explicit equations for the state vector, it is even not clear to us which mathematical result should be invoked to rigorously guarantee convergence and convergence time estimates for a generic system. This is at stark variance with methods based on ordinary stochastic differential equations. Furthermore, the need to keep track of distinct realizations hints the that the method of [30] may be more memory and computing power demanding. We want to make clear that we are not disputing the validity of the algorithm of [30], but in the absence of a rigorous mathematical formulation of [30] a precise assessment remains difficult.

3. Are there any new insights that this method brings?

- There are new insights that we now discuss in the revised version. First, in the revised version of the paper, Methods section, we present a general framework to construct the influence martingale.

The unraveling of completely-positive-divisible maps differs from positive-divisible or just divisible maps only in consequence of the sign of weights that enter the Monte Carlo averages specifying the state operator. In other words, the result of the quantum average embodied by the partial trace which specifies the state vector can be replicated by a classical average formally extended to "events" with "negative probability". The occurrence of such "events" hinges upon the degree of positive-divisibility of the dynamical map. In the revised version we highlight the correspondence of our findings with Feynman's discussion of the double slit experiment in ref [44] and [45]. There Feynman shows how averages with respect to a genuine quantum statistics (as the partial trace in our case) can be reproduced by classical averages formally extended to "events" weighed by "negative probability" (as the influence martingale effectively does in pathspace). Feynman evinces how the use of "negative probability" as an intermediate computational step does not cause any logical difficulty. Namely, Feynman interprets "negative probabilities" as the hallmark of events which either cannot be directly verified or that cannot be simultaneously observed. Feynman further argues that observation of interference patterns in the double slit experiment depends not only upon the fact that the two slits are open but also on the positioning of detectors. In particular, he argues that if detectors were placed before each slit, interference patterns would disappear. It seems to us that a similar reasoning applied to quantum trajectories suggests that a continuous-time-measurement-interpretation of quantum trajectories may be always possible but would correspond to placing detectors on the slits, thus effectively suppressing interference effects. In other words, the degree of positive divisibility introduces a further element of contextual dependence on the "subjective reality" of the trajectories.

Overall our results call for a careful reassessment of the continuous time measurement interpretation of quantum trajectories. The claim made in [32] that no measurement interpretation is possible because "*the jump probabilities and operators connect different trajectories, in a way that the event at a time t on a given trajectory will depend on the previous events also on all the other trajectories.*" appears instead only based on non necessary conditions imposed by the unraveling algorithm [30].

Further insights consists in the possibility offered by the influence martingale method to unravel hybrid (classical quantum) master equations as already mentioned. We are thus able to derive exact arbitrary coupling quantum trajectory pictures of calorimetric measurement. This fact validates and extends previous results obtained in the weak coupling regime.

Turning to the specific questions

1. In Eq. 1, I would state specifically that the Gamma can be time-dependent.
 - We did it before equation two.
2. In the discussion of P3, first column - this is reliant on looking at the other papers referenced - it is not possible to follow what the authors have done, even as an expert,

without reference to these other papers. I therefore think the authors need to put in more detail here.

- We now give more details, taking into account space constraints.
3. In Figs. 2 and 4 - though the agreement does seem to be good between the new method and the master equation there does, nonetheless, seem to be some systematic disagreement. Have the authors checked that the results will eventually converge? If they are not exactly converging - why is this?

- To start with, the convergence of the results is guaranteed a proven mathematical fact. In the first version of the paper we repeated our numerical integrations up to ensembles of $O(10^4)$. One order of magnitude less of the ensemble size used in ref [30]. In all cases we noticed systematic agreement between Monte Carlo averages and direct integration of the master equation in the following mathematical sense: 1) the numerical values are systematically within error bars estimated by fluctuations of the Monte Carlo averages (round-off errors are too small to be significant for our examples) 2) fluctuations decrease in magnitude as the ensemble size increases as expected from the theory.

We have remade the figures to clearly show the mathematically proven convergence. Concretely, the original Figure 2 has been split up in two figures. The new Figure 2 now contains overlapping curves of 10^4 and 10^5 realisations, the reduction in fluctuations clearly displays convergence. The new Figure 3 shows that also the trace of the density matrix is well-behaved. We also added Figure 5 which shows for a different example a comparison of 10^3 and 10^4 realisations, again illustrating the convergence. It seems to us that the numerical results are in perfect agreement with what the theory predicts. In particular the theory predicts convergence in any finite time interval but not necessarily uniform convergence. Therefore is not surprising that the size of fluctuations may grow for a fixed ensemble as time increases. This is a problem common to any numerics involving exponentially growing indicators. In the literature see e.g. [47] suitable algorithms have been developed to milden the problem.

4. P3, col 2: the authors discuss information back flow - can they plot a measure of this, so that it can be verified for this case?

- We refer to the explicit formula (66) of [50] and explain in words that it "relates the direction of the information flow in the model to the sign of the time derivative of $|\Gamma_t|$ ". Ref [50] contains a thorough discussion of the information back-flow which is thereby identified as an indicator of non-Markovianity. As the rules of NATCOMMS put limits to the number of plots we prefer to reserve more space for illustrating our new results.

5. P4. In the last sentence of the conclusion, the authors say that the influence martingale yields a significant contribution to the first two of their three interpretations. Can the authors expand on this, especially on how it does this for their second interpretation?

- We expanded the conclusions. We discuss the relation pointed out by Feynman between interference and "negative probability", see above the reply to the "new insights" question.

Reply to Reviewer #2

We warmly thank the reviewer for the careful reading of our paper. Indeed the report summarizes our results even more effectively than we were ourselves able! Concerning the remarks

1. If I am not missing something, the quantum trajectories defined by Eqs.(3)-(5) would be able to reproduce an evolution of the average state as fixed an equation (1) even if such an evolution is not completely positive (and actually not even positive) ...
 - Indeed the unraveling works for any just divisible quantum map. We state this fact more explicitly. Furthermore we include among the examples the unraveling of a master equation (Redfield equation) which does not preserve positivity.
2. In a way related with the previous point, before Eq.(2), instead of saying that the Lindblad master equation follows from the postulate of quantum mechanics if and only if Eq.(2) hold, I would rather mention explicitly the requirement of complete positivity of the dynamics as well as of its propagators (one might even directly mention the property of CP-divisibility, which is quite well-established in the literature).
 - We now do so.
3. In the second column of page 1, 10 lines from the end, please note that “introduction an extra” should read “introduction of an extra”.
 - Thanks. We have entirely rewritten that paragraph with the aim of responding to the points raised by Reviewer #3
4. In the caption of Fig.1, shouldn't μ_t be referred simply to the solid line, rather than to the x-shapes?
 - Thanks, we have fixed the caption.
5. It might be convenient to include the description of the symbols also in the caption of Fig.2.
 - The captions now include a description of the symbols.

Reply to Reviewer #3

1. Let me start by saying that while I work on quantum stochastic processes, I do not work on quantum trajectories. ... Overall, I think the paper is not easy to read, even for an expert in quantum stochastic processes. I think the contributions may be significant, but I was unable to ascertain this from the paper itself.
 - We thank the reviewer for this frank statement. We believe that our result is significant for two reasons. First, because it proves the existence of a quantum trajectory picture for all divisible quantum maps. Most importantly, quantum trajectories take values in the Hilbert space of the system and are always Markov processes defined by the solution of an ordinary stochastic differential equation. The second reason is that the proof of our result, which is rigorous, only requires a very elementary tool: Itô lemma. We took therefore very seriously the comment that the paper is hard to read. We have completely rewritten the central part of the paper in order to emphasize the central idea of the paper: extension of the well known Girsanov formula in the theory of Markov processes. We now recall what is Girsanov formula

and in which sense we do extend it.

2. I do not see how Eq. 5 is justified. It's not derived in the Methods section.

- We expanded the text around the equation that we require the process μ_t to satisfy. We explain that the equation is not derived but assigned based on the requirement that its solution μ_t enjoys the martingale property with respect to the Poisson process.

3. I am not sure what is happening in the Methods section as it relies on Eqs 3,4,5.

- We have rewritten the main text to make clear the logic. We start from the Ansatz that the state operator can be computed as a Monte Carlo average formally equivalent to Girsanov formula. The difference with Girsanov formula is that the martingale is allowed to take negative values. Next we assign the equation for the state vector (we essentially choose the same equation as Dalibard Castin & Møllmer) and the equation for the martingale. We then apply Itô lemma. The calculation is done step-by-step in Methods. There we show that our choices guarantee that the Monte Carlo average defining the state operator indeed satisfies the master equation. In Methods we added a new section where we repeat the construction in a slightly more general fashion in order to highlight the consistence condition that the parameters in the assigned dynamical equations for state vector and influence martingale must satisfy in order to unravel the master equation. We emphasize that unravelings are not unique exactly because they at most correspond to a "subjectively" (contextually) real description of a physical process measured by a specific setup.

4. I am not convinced of the "interference" interpretation of the slave master equation. Afterall, it's just adjusting weights. Often quantum states must be decomposed with affine weights, but it may not mean interference. Can they provide more physical argument? I am also skeptical since quantum trajectories themselves are not physically real in a sense

- This was a really fruitful objection. It allowed P.M.-G. to recall an evening spent in the library of Niels Bohr Institute in Copenhagen during his Ph.D. studies reading Feynman's essay "negative probability", ref [44] in the manuscript. Feynman there shows that averages over a genuine quantum statistics can be reproduced by a classical Monte Carlo average by adjusting weights, as we do, in order to formally attribute certain physically non directly verifiable "events" a "negative probability", as we do. Feynman analyzes in particular the double slit experiment and shows that exactly the cases when interference patterns appear require adjusting weights with negative values. Feynman's argument corroborates, if not via unconscious memory, inspired our interpretation. Paths contributing to path integral partial trace do interfere. We are reproducing this effect by means of a classical average. Completely positive divisible maps are generically constructed from microscopic models at weak coupling. In that case quantum effects in the statistics are negligible. The influence martingale is positive definite and can be reabsorbed in the definition of the Poisson process (see discussion in the revised section Methods). In Feynman's double slit example this situation corresponds to closing one slit so that the statistics of counted detection is peaked around the open slit. When the quantum map is only positive divisible or just divisible, negative weights appear. Quantum maps of this sort appear at strong coupling (exactly integrable models, master equations generated by TCL perturbation theory). The partial trace defining the state operator is now af-

ected by quantum effects and we need negative weights, exactly as in Feynman's discussion of the full fledged double slit experiment. In summary, we adjust weights to reproduce a quantum statistics which we describe as producing interference in quantum trajectory path-space. Is this the only possible interpretation? Possibly not, but we do not see any simpler physical picture.

5. The Numerics section of the paper was really difficult to penetrate. I am sure the numerics are all fine, but what should I get out of this? The authors don't seem to compare to previous methods, which presumably fail.

- We now introduce the numerics section with a paragraph where we illustrate its purpose. In particular we emphasize that 1) the influence martingale method permits to apply the well known theory of ordinary stochastic differential equations to the unraveling of divisible dynamical maps. Hence one has at hand the bounty of choosing among many existing algorithms to efficiently implement our theoretical result depending to specific exigences. 2) We consider examples 3 and 4 that the influence martingale is able to handle whilst other methods are not. This is due to the fact that the influence martingale only imposes integrability requirements on the coefficients $\Gamma_{\ell,t}$ of the master equation but does not impose any positivity requirements.

6. How does the method compare to exact calculations?

- Concerning analytic results, our main result provides a general way to unravel time local master equations. The result holds independently of the fact that the equations are exactly integrable or not. The point is that we prove which stochastic differential equations need to be integrated if one is after a quantum trajectory picture of a divisible dynamical map. Before, to the best of our knowledge, it was not known. Concerning comparison with direct integration of the master equation, quantum trajectory pictures have an edge for systems with a large number of states. In our examples our goal was only to show the flexibility of the method and therefore to compare it with direct integration of the master equation when the latter is feasible. We have remade Figures 2, 3 and introduced new Figures 5 and 6. In all of these figures the full black lines indicate the solution by directly integrating the master equation. The diamonds, crosses and x's indicate predictions by the influence martingale method and the shaded area shows the fluctuations. The numerical solution of the master equation always falls well within the shaded regions, indication good correspondence with the Monte Carlo simulations.

7. It's not clear how difficult are problems that they do consider for numerics.

- Our goal in the numerics section is to visually exemplify the main result including cases to which a quantum trajectory picture in the Hilbert space of the system cannot be associated by other means. We looked for the simplest examples having the specific properties we wanted to illustrate. Algorithmic implementations of our main result to cases where a quantum trajectory picture in itself has an edge with respect to direct numerical integration of the master equation are beyond the scopes of our work exactly as they were for the rightly celebrated ref. [16] or for ref. [30].

Sincerely Yours,

Brecht Donvil,

Paolo Muratore-Ginanneschi

Reviewers' comments:

Reviewer #1 (Remarks to the Author):

The authors have indeed significantly revised and improved the paper, now discussing the derivation of the influence martingale result and the interpretation in more detail. They have also been able to show the results they present converging in a convincing way.

However, I am not satisfied that they have shown any new calculation that this method opens that could not straightforwardly be done using other techniques. Indeed, they compare their quantum trajectory results with master equation results throughout the paper. I asked previously about how the time taken for their method to compute dynamics compares to others. They have not done any concrete, numerical studies of this question - and indeed, I would imagine that for the examples shown the master equation predictions would have been quicker to obtain than the converged trajectory simulations.

When I discussed other methods in my previous report, I was not only referring to other stochastic trajectory techniques - I would also include other methods for simulating non-Markovian dynamics (e.g. TEMPO, HOPS, TEDOPA, etc). The authors have not engaged with this work. I also think the "influence martingale" term, which is inspired by Feynman's influence functional, is somewhat misleading - the influence functional is a powerful tool which ultimately allows the exact simulation of an open system linearly coupled to a harmonic environment. This paper is not based on such a microscopic foundation, rather here the authors investigate abstract models or reproduce results from other techniques (e.g. the Redfield equation).

The authors claim in a discussion that they are preparing a paper including a calculation that could not be done by other methods. In my view, a calculation that does this needs to appear in this paper in order for it to be considered for Nature Communications.

Reviewer #3 (Remarks to the Author):

Firstly, apologies for my late report. The Manuscript has changed significantly and it took me some time to get through it. I do think that the presentation is much better now. I am still not fully convinced by the results. It is possible that my concerns are due to my own lack of understanding of quantum trajectories. However, it would be good if the authors address this matter directly in the paper.

The authors use rather vague arguments to justify negative rates. They rely on an essay by Feynman, which is nice, but it's a bit dated. I would have liked to see an argument that for a classical master equation, i.e., one with no coherences, there can never be negative rates. If this is the case then I am inclined to believe that this is a genuine quantum feature.

It is true that a stochastic matrix is always positive and therefore I can envisage a fully positive classical master equation, even when a process is non-Markovian. However, the reasons why not completely positive maps exist has to do with non-operational considerations that are dubious at best. This worries me about the validity of a master equation and the corresponding stochastic trajectories.

My understanding of stochastic trajectories is that they correspond to a sequence of Kraus operators. Now consider the cases where time local master equations are constructed using NCP maps, e.g. arXiv:1009.0845. Where lies the problem in constructing stochastic trajectories here?

Or even better, consider the arXiv:1307.7743, and subsequently arXiv:1704.06204, where non-Markovian master equations are given in terms of a family of CPTP maps. Each of these can then be written in terms of Kraus operator, which then yield trajectories. In this case, the weights in front of these trajectories will not be positive, but the dynamics are guaranteed to be faithful.

I am not convinced that the same holds for the trajectories given in the paper. Although, the authors claim that they do. Perhaps, the claim is that given any master equation they can provide stochastic trajectories, without caring for the physicality of the master equation.

So, my concern then boils down to be the same as Referee 1; how does this add to what is known, e.g. arXiv:1307.7743?

On the presentation and the technical side, I didn't understand the intuition for the martingale property. Is it effectively a trace preservation condition? I also don't understand why is (4) Poisson processes?

Reply to Reviewer # 1

We thank the Reviewer for taking the time to assess our manuscript and for providing further comments.

Our overall impression of the report by the reviewer is that it overlooks the theoretical background of our result. We have rewritten the introduction of the paper to more clearly motivate the significance of proving that any time local master equation admits an unraveling in term of a Markov process in the Hilbert space of the system.

Let us briefly summarise our main points also here.

- We prove that any divisible quantum map admits a non-anticipating (i.e. causal) unraveling in the Hilbert space of the system. Our result is thus a radical change of perspective with respect to claims, most prominently represented in the literature and highly quoted, that the unraveling of positive divisible or just divisible maps requires non-Markovian features (anticipating correlations or enlargements of the Hilbert space). We show instead that the same unraveling theory handles divisible maps on the exactly same footing as completely positive divisible.
- Non-anticipating unravelings are needed to address questions like the existence of a measurement interpretation, quantum state prediction and retrodiction and quantum state parameter estimate based on **individual sequences of measurements**. Our result thus overcomes the main objection raised e.g. by Wiseman and Gambetta in PRL 101 140401 (2008) against the measurement interpretation of the Diosi-Gisin-Strunz quantum trajectory theory. Furthermore, this latter theory only holds for the restrictive case of a linearly coupled Gaussian environment. Our result holds generically whenever the **effective potential** (i.e. the counterpart of the influence functional generated by a linearly coupled Gaussian environment) is handled by time-convolutionless methods.
- Our result paves the way to develop new numerical algorithms for integrating open quantum systems in **high dimensional Hilbert spaces**. It is well-known that in high dimensional Hilbert spaces algorithms based on the stochastic Schrödinger equation provide numerical advantages over integrating the master equation directly. In the current version we recall the theoretical arguments and exemplify them in a concrete case.

Without changing the core result, we have made several additions to the paper that illustrate the above points.

1. We discuss the relation between general time local master equations and completely bounded maps and the Wittstock-Paulsen decomposition. The Wittstock-Paulsen decomposition states that any completely bounded map can be written as the difference of two completely positive maps. We explicitly construct the decomposition using the influence martingale and show that the sign difference between the maps is directly related to the sign of the martingale.
2. We added a new application (Applications: Simulating Large Quantum Systems). There we explicitly reproduce theoretical argument explaining why unravelings offer an advantage over direct integration of the master equation for **large systems**. As an example, we study the dynamics of a spin chain and find that as the length of the spin chain

increases, the computation time for the unraveling based on the influence martingale becomes smaller than directly integrating the master equation (Fig. 6).

3. We show how photocurrent oscillations can be simply accounted by microscopic models giving raise to a positive divisible dynamics for the system. The photocurrent is defined in terms of a jump process (see e.g. chapter 4 of Wiseman and Millburn pag 155). The calculation, although very simple, is trivially impossible without unraveling because without unraveling there is no average over a jump process i.e. over sequences of detected events.

We now come to a point by point reply to the criticism.

Reviewer *However, I am not satisfied that they have shown any new calculation that this method opens that could not straightforwardly be done using other techniques. Indeed, they compare their quantum trajectory results with master equation results throughout the paper. I asked previously about how the time taken for their method to compute dynamics compares to others. They have not done any concrete, numerical studies of this question - and indeed, I would imagine that for the examples shown the master equation predictions would have been quicker to obtain than the converged trajectory simulations.*

As we now make clear in the revised version, the scope of discussing **low dimensional systems** in the example section is exclusively to show in the simplest possible cases how **physical phenomena** such as e.g. revivals are well captured by a Markov process in the Hilbert space of the system. Previously these phenomena were thought to be hallmarks of non-Markovian (in the sense of correlation with events in the past and in the future – memory/prescience) dynamics. In particular one example shows the existence of Markov quantum trajectories in the Hilbert space of the system in a context where it was recently argued to be impossible (Smirne et al PRL 124, 190402 (2020)).

Large dimensional systems are the natural context were our theory offers computational advantages. The reason is the same as for the case of competely positive divisible maps. For the Reviewer convenience we reproduce here Wiseman, H. M. and Milburn, G. J. “Quantum Measurement and Control” Cambridge University Press, 2009 478 beginning of page 153.

The advantage of doing [quantum trajectory] rather than solving the master equation is that, if the system requires a Hilbert space of dimension N in order to be represented accurately, then in general storing the state matrix ρ requires of order N^2 real numbers, whereas storing the state vector $|\psi\rangle$ requires only of order N . For large N , the time taken to compute the evolution of the state matrix via the master equation scales as N^4 , whereas the time taken to compute the ensemble of state vectors via the quantum trajectory scales as N^2M , or just N^2 if parallel processors are available. Even though one requires $M \gg 1$, reasonable results may be obtainable with $M \ll N^2$. For extremely large N it may be impossible even to store the state matrix on most computers. In this case the quantum trajectory method may still be useful, if one wishes to calculate only certain system averages, rather than the entire state matrix

(The same argument was previously put forward by Dalibard-Castain-Møllmer PRL1992).

Reviewer *When I discussed other methods in my previous report, I was not only referring to other stochastic trajectory techniques - I would also include other methods for simulating non-Markovian dynamics (e.g. TEMPO, HOPS, TEDOPA, etc). The authors have not engaged with this work. How does the computation time for this method compare to other techniques?*

In the new version of the paper we have added a new example to address computation time of the unraveling based on the influence martingale and integrating the master equation directly as the **dimension of the system increases**. We show that for a system of N spins the theoretically expected crossover of the computing time curves occurs already for N=10.

In our previous reply to the reviewer, we already discussed how the numerical performance of previously known unravelings is burdened either by memory/prescience effects or the need to introduce tensor product with auxiliary Hilbert spaces.

We now briefly consider the algorithms mentioned by the Reviewer. We want in any case to clearly state that our comments are not criticisms to the papers we mention.

1. TEMPO as described in Strathearn et al in NATCOMMS (2018)9:3322. The authors work “in a representation where $d \times d$ density operators [state operators in our language] are given instead by vectors with d^2 elements”. The argument of Dalibard-Castain-Møllmer thus applies. In fact, the main example considered is a **single spin**. The method seems to be able to handle well systems linearly coupled to Gaussian environments, although no clear efficiency comparison is provided. We do not need such simplifying hypothesis (see the next point).
2. HOPS (Suess et al PRL 113, 150403 (2014)): the quantum state diffusion of Strunz Gisin and Diosi is a stochastic differential equation with memory/prescience derived for Gaussian environments linearly coupled to a system. In general there is no time local master equation in this case. Of course approximate time local master equations can be constructed for Gaussian environments linearly coupled to non-linear systems via time-convolutionless perturbation theory (TCL). But TCL also applies to non Gaussian environments. Our theory applies to time local master equations derived from microscopic models via TCL or in examples when the exact evaluation of the partial trace on a microscopic model is possible (photonic band gap, Gaussian environment with Gaussian system, central Fermion model). The domains of application of our theory and HOPS have some overlap but do not coincide.
3. TEDOPA (as of Prior et al PRL 105, 050404 (2010)) the method is applied for **two spins** linearly coupled to a Gaussian environment. Furthermore the method relies the time-adaptive density matrix renormalization group (t-DMRG) technique. As Schollwöck states in the abstract of the highly quoted Rev Mod Phys 77, 2005, DMRG is adapted to low-dimensional systems.

In conclusion, simple theoretical considerations show that **scalability** and **Markovianity** are the advantage offered by our methods. We would like to draw the attention of the Reviewer that these are the same reasons for resorting to Lagrangian algorithms (i.e. integration via stochastic differential equations) instead of Eulerian (integration via partial differential equations) in classical hydrodynamics.

Reviewer: *I also think the “influence martingale” term, which is inspired by Feynman’s in-*

fluence functional, is somewhat misleading - the influence functional is a powerful tool which ultimately allows the exact simulation of an open system linearly coupled to a harmonic environment. This paper is not based on such a microscopic foundation, rather here the authors investigate abstract models or reproduce results from other techniques (e.g. the Redfield equation).

To start with, we list in the paper the microscopic models for which the partial trace can be exactly computed and give raise to a time local master equation of the form we consider. The photonic band gap of our first example is one of them. The photonic band gap is taken from a highly influential paper by John and Quang [33] whose microscopic foundation is undeniable. In any case, exactly integrable environment models are exceptional and their interest has to be justified case by case.

For instance, a Gaussian model linearly coupled to the system has historical and physical interest (Leggett's 1983 argument) for thermal baths (infinite environment with continuous spectrum) which relax to equilibrium much faster than the system. But as well explained in the paper that introduced TEPODA Prior et al PRL 105, 050404 (2010)

assuming that the correlation time of the environments is much faster than the system dynamics is frequently not justified in many realistic systems.

From the point of view of modern quantum and statistical field theory, the influence functional of Feynman and Vernon is an exactly integrable example of what is generally called an effective potential.

As we make clear in the revised version, our theory applies to the **universal** (and not abstract! we are not resorting to any postulate) consequences of integrating out the degrees of freedom specifying the effective potential.

In the language of renormalization group theory, a property is universal if it holds for the coarse-grained system provided that conservation laws (the state operator trace for us) are preserved.

Time convolutionless perturbation theory is a form of renormalised perturbation theory. It allows one to derive time local master equations at any order in the system-environment non-dimensional coupling constant **for any microscopic dynamics with unitary evolution admitting an expansion**. See for details chapter 11 of Breuer and Petruccione or the refs we added to the revised version.

Finally, we included the Redfield model to show that our result **extends** the domain of application of unraveling theory by Markov processes even in cases when this was deemed impossible. **Reviewer # 2 explicitly asked us to emphasize this point**. Redfield models are widely used in applications. In fact, we have added references [21-26] to more clearly motivate the use of non-positivity preserving maps.

Reviewer: *The authors claim in a discussion that they are preparing a paper including a calculation that could not be done by other methods. In my view, a calculation that does this needs to appear in this paper in order for it to be considered for Nature Communications.*

The computation hinted at in the discussion is too complex and involved to condense into the current manuscript. Therefore, we removed the reference to it in the new version of the

manuscript. In order to display the strength of the method, we decided to describe how it is possible to derive photocurrent oscillations from time-convolutionless master equation see point 3. (on page 1).

We thus replied to all the criticisms raised by the reviewer. In particular we provided concrete examples of theoretical (photocurrent oscillations, mapping to an instrument) and numerical (efficient integration of divisible dynamics in high dimensions) applications that only are possible based on our result.

Reply to Reviewer # 3

We thank the Reviewer for taking the time to assess our manuscript and for providing further comments.

Reviewer *It is possible that my concerns are due to my own lack of understanding of quantum trajectories. However, it would be good if the authors address this matter directly in the paper.*

We re-wrote our introduction to unambiguously explain what we mean by quantum trajectories (i.e. unraveling in terms of a stochastic process) and why quantum trajectories are interesting for quantum measurement theory.

Quantum measurement theory is our main interest. The purpose of the numerical examples is to show how a Markovian unraveling in the Hilbert space of the system is able to capture physical phenomena previously believed to be related to memory and prescience. From the numerical point of view, it is known since the Dalibard Castain Mølmer paper (PRL 1992, 68, 580, see also. chapter 7 of Breuer Petruccione or chapter 4 of Wiseman and Millburn) that quantum trajectories (=unravelings) offer an advantage for high dimensional open quantum systems. See the point by point reply for details.

We now come to a point by point reply to the criticisms of the Reviewer.

Reviewer: *Perhaps, the claim is that given any master equation they can provide stochastic trajectories, without caring for the physicality of the master equation.*

It is not clear to us what is meant by "the physicality" of the master equation.

In the revised introduction we now refer to the works of Pechukas, Sudarshan and most recently Lidar [21-26] where the authors rigorously prove the physics interest of non completely positive dynamics starting from the von Neumann postulates.

If the Reviewer means that we consider the Redfield example which violates positivity preservation, we want to note that Reviewer # 2 explicitly asked us to emphasise this point.

Reviewer: *The authors use rather vague arguments to justify negative rates.*

If by negative rates the reviewer means what we call negative Lindblad weights, these quantities do not need to be justified, neither our argument is aimed at justifying them. Negative values of Lindblad weights are a mathematical fact which can be explicitly derived from microscopic models sometimes even exactly. Of course, it is possible to interpret the physical phenomena which occasion them. In order to dispel any possible confusion, in the revised version of the paper we recall that the occurrence of negative weights (which are probabilities per unit of time) is a consequence Khalfin's theorem. As well known, Khalfin's theorem forbids asymptotic exponential decay of survival probabilities in unitary dynamics with continuous spectra. Outside the intermediate asymptotic specified by weak coupling, time derivatives of transition probabilities (Lindblad weights) are therefore not constrained to be positive definite. We also add the reference to Fonda, Ghirardi and Rimini Report on Progress on Physics 41, 587 (1978) where the phenomenon is clearly explained as rescattering (a form of interference) from the environment to the system.

We refer to Feynman's essay to interpret negative values of the influence martingale in the unraveling of the master equation. A simple calculation shows that negative values of the weights

are a necessary but not sufficient condition for negative values of the influence martingale. Therefore the interpretation of negative values of the martingales is a distinct issue.

Drawing from Feynman, the point we tried to make in previous versions of the manuscript is that a quantity with a probabilistic interpretation (a partial trace for a positive divisible dynamics) comes about as the difference rather than the sum of positive definite contributions.

In the revised version of the manuscript we also provide also an operator algebra explanation of the fact. The influence martingale weighs the classical statistical average so to reproduce the canonical form of a completely bounded map. Theorems by Wittstock and Paulsen prove that completely bounded maps admit an algebraic characterisation as the difference of completely positive maps. Positive maps are a subset of completely bounded maps whose algebraic characterisation in general is not known.

Reviewer: *They rely on an essay by Feynman, which is nice, but it's a bit dated.*

We have added refs Abramsky and Brandenburger (2011), (2014) Blass and Gurevich (2021) (refs [45-47] in the manuscript) where Feynman's argument is the starting point to give a mathematically rigorous operational meaning to negative probability. Such operational characterization is well adapted to describe the statistical characterization of positive divisible maps which emerges from our unraveling. The essay by Feynman thus still motivates research by prominent figures of the mathematical logic and mathematical physics communities.

Reviewer: *I would have liked to see an argument that for a classical master equation, i.e., one with no coherences, there can never be negative rates. If this is the case then I am inclined to believe that this is a genuine quantum feature.*

A classical master equation is an equation governing the evolution of the probability distribution of the values taken by a stochastic process **describing an observable** at any instant of time. Therefore the master equation is specified by the generator of the stochastic process. Rates are necessarily positive because they are specified by the quadratic variation of the process.

Furthermore, exact coarse graining of a classical probability distribution means summing or integrating over positive terms. No cancellations and therefore interference are possible.

The probabilistic content of the solution of a quantum master equation is not the probability distribution of the realizations of the stochastic process (the stochastic state vector or if the Reviewer prefers the wave function) which unravels it. Thus there is no constraint on the sign of the dissipator unless the dynamics is completely positive divisible (i.e. in the Lindblad-Gorini-Kossakowski-Sudarshan case). The Lindblad-Gorini-Kossakowski-Sudarshan master equation is usually derived in the weak coupling scaling limit which corresponds to the first order of a time convolutionless perturbation theory. Higher orders mean stronger coupling and further quantum effects included in the partial trace. The partial trace is a sum over complex numbers with no sign constraints. This is the phenomenon the influence martingale is modeling and which we interpret as interference.

Finally, a direct extension of Teich and Mahler's argument (PRA 45 3300 1992) proves that any time local master equation can be formally rewritten as Pauli master equation without coherences and with "rates" necessarily positive in Lindblad-Gorini-Kossakowski-Sudarshan case. Again, absence of coherences does not imply any classical interpretation. Furthermore the resulting unraveling is numerically costly and bereft of measurement interpretation (Wiseman

and Toombes PRA 60 1999).

Reviewer: *It is true that a stochastic matrix is always positive and therefore I can envisage a fully positive classical master equation, even when a process is non-Markovian.*

A. Shaji and E.C.G. Sudarshan argued clearly in their work Physics Letters A 341 (2005) 48–54 for the use of non-completely positive maps. We quote from their conclusions

We find that the arguments that are put forward to often justify considering only completely positive maps as possible descriptions of open quantum evolution do not stand up to closer inspection.

and then

It must be emphasized here that once the requirement of complete positivity is relaxed then the positivity of the reduced dynamics has no special significance. Initial correlations with the environment can lead to open quantum dynamics that can be not positive also.

Reviewer: *However, the reasons why not completely positive maps exist has to do with non-operational considerations that are dubious at best. This worries me about the validity of a master equation and the corresponding stochastic trajectories.*

We have added references to the introduction which clearly motivate the use on not completely positive maps, see Pechukas, Sudarshan, Lidar and from the phenomenological point of view Hartmann and Strunz (refs [21-26] in the manuscript).

Furthermore, a careful distinction should be made between (non-)completely positive and (non-)completely **divisible** maps. Exactly integrable models like Gaussian system environment models (e.i. boson-boson or fermion-fermion) can be non-completely positive divisible while their evolution is still a completely positive map.

Reviewer: *"Now consider the cases where time local master equations are constructed using NCP maps, e.g. arXiv:1009.0845. Where lies the problem in constructing stochastic trajectories here? "*

The problem lies in the fact that to the best of our knowledge an unraveling with the desirable feature of a state vector being a Markov process in the Hilbert space of the system did not exist before our discovery. In one row: all master equations described by a semi-group have a non-anticipative unraveling.

Furthermore, arXiv:1009.0845 explores the dynamics of non-completely positive dynamics and looks for characterisation of non-Markovianity. Indeed, when arXiv:1009.0845 appeared the authors could only refer to their ref [19] (Piilo et. al PRL 2008 [37] in the manuscript) which upholds the necessity in the Hilbert space of the system of complicated memory/prescience correlations for the unraveling in quantum trajectories of the dynamical maps. As said, our results show that a radically different, Markovian, picture holds true.

Reviewer *Or even better, consider the arXiv:1307.7743, and subsequently arXiv:1704.06204, where non-Markovian master equations are given in terms of a family of CPTP maps. Each of these can then be written in terms of Kraus operator, which then yield trajectories. In this case,*

the weights in front of these trajectories will not be positive, but the dynamics are guaranteed to be faithful.

Tensor networks and unravelings are conceptually different approaches adapted to address different type of questions (see more below).

Reviewer: *So, my concern then boils down to be the same as Referee # 1; how does this add to what is known, e.g. arXiv:1307.7743?*

Reviewer 1 in the second report says that the main concern is the lack of a comparative study of numerical efficiency with direct integration of the master equation.

To make clear the theoretical context, we quote the introduction to chapter 7 of the celebrated monograph by Breuer Petruccione on Open Quantum systems (a similar paragraph can be found in Wiseman and Millbourn 2009 pag 153)

To determine numerically the density matrix of an open quantum system one can either integrate the density matrix equation directly or else simulate the process for the stochastic wave function and estimate the covariance matrix. In general, the density matrix equation leads to a system of linear equations involving N^2 complex variables, N denoting the effective dimension of the Hilbert space. By contrast, stochastic simulation only requires the treatment of N complex variables characterizing the state vector. For large N , that is for high-dimensional Hilbert spaces, one thus expects Monte Carlo simulations to be numerically more efficient than the integration of the density matrix equation, provided the size of the required sample of realizations does not increase too strongly with N .

In the revised version we now show how for large systems, the unraveling relying on the martingale does indeed become numerically more efficient than integrating the master equation (see Fig. 6). We do this by studying a chain of interacting spins and compute the state operator by direct integration and unraveling using the well-established QuTiP package for Python.

arXiv:1307.7743 (as any tensor network) works with the state operator. Breuer and Petruccione's argument on numerical efficiency applies if one tries to deploy such method to systems with a large number of states. This observation must not be read as a criticism to arXiv:1307.7743. It is only meant to clarify why numerical application of our results serve other purposes (as Reviewer 2 recognised). In one word scalability is the point.

Reviewer: *On the presentation and the technical side, I didn't understand the intuition for the martingale property. Is it effectively a trace preservation condition?*

Indeed, if you want the outer product to be defined on the Bloch hypersphere for individual realizations of the unraveling the scalar prefactor has to be a martingale. The preservation of the Bloch hypersphere is a central point in the paper and for the interpretation of quantum trajectory theory, it is discussed throughout the paper. As we now point out in Discussion, the martingale condition can be relinquished at the price of trace preservation only on average.

Reviewer: *I also don't understand why is (4) Poisson processes?*

We do not say that (4) is a Poisson process. We say that (4) is a stochastic differential equation driven by Poisson processes. Equation (4) is in the form of the original jump equation

introduced by Dalibard Castain Mølmer (when the weights are all positive definite it is). In this equation, the Poisson processes model quantum jumps. We extend the quantum jump approach to all trace preserving time local master equations.

REVIEWER COMMENTS

Reviewer #1 (Remarks to the Author):

The authors have written a detailed response to my previous points. Importantly, they have now provided a comparison of their trajectory method with direct integration of the master equation and shown where the method becomes more efficient.

Assuming this calculation could not have been done with other trajectory methods, then I am happy to recommend publication.

Reviewer #3 (Remarks to the Author):

I really appreciate the authors spending a great deal of time revising this manuscript and responding to my previous report in great detail. There are several things in their response that simply do not agree with. I will start with these issues:

The authors' response to my comment "classical master equations" and "negative rates" is not answered qualitatively enough. After reading their comment I am still not convinced that if one takes a model with the density matrix is always diagonal then the corresponding influence martingale will be positive. While I agree that there are interference effects, the memory could reduce the probability of an event, when the process is non-Markovian. The current argument simply motivates NCP superoperators as differences in CP maps. This is fine, but this is not the same as saying that such dynamics must be quantum. I don't think this will be true. If it were, this would be as strong as the nonpositivity of the Wigner function of the Sudarshan-Glauber p-function.

Their reliance on Shaji-Sudarshan is rather misguided. The program of not-completely positive (NCP) quantum maps is highly ambiguous. These maps cannot be observed experimentally as their input states are not well-defined. The Shaji-Sudarshan (along with Jordan) present mathematical maps that are fine, but physically meaningless. This is precisely what I worry about with the current manuscript. However, one can interpret a non-Markovian master equation as differences in different CP stochastic maps. This is exactly what transfer tensor does. The authors make a claim that the transfer tensor is a "tensor network" method that has to track a density matrix. This is not true. It's not a tensor network method, and one can simply compute a family of trajectories of a pure state and then sum them up. This should be identical to the method they propose here.

Despite these rather important issues, I found the latest version of the manuscript much more approachable.

Issue 1 is still important foundational but does not challenge the validity of the method. However, I do think it should be addressed in the paper a bit more formally. The current claims seem over the top; either prove that nonpositivity must be quantum or don't say it. They should soften the claim.

Issue 2 is not important, because if one derives a master equation from total Hamiltonian dynamics, and makes sure that the reduced state of the system is consistent with the total dynamics then that master equation is physically motivated. This is different from what Shaji-Sudarshan wanted to do; they were worried about quantum process tomography and NCP dynamics. This problem is better solved using the language of instruments (see Milz-Modi PRX Quantum 2021). Importantly, the authors show that their method gives the same results as the master equation.

Issue 3 does not affect their presentation so we can overlook this matter.

I have one final issue. On top of page 8 it reads:
"We need therefore to apply Ito lemma to evaluate

$d(\mu_t \psi_t \psi_t^\dagger) = (d\mu_t) \psi_t \psi_t^\dagger + \mu_t d(\psi_t \psi_t^\dagger) + (d\mu_t) d(\psi_t \psi_t^\dagger)$
in the case of Poisson noise.”

I still don't understand the Poisson noise issue. This seems like a big assumption. Can the authors argue that this does not take away from the generality of their results? My feeling is that the wording is the issue, not the derivation. If this is indeed the case, I will recommend publication.

Minor comments: The usage of quotations is off throughout the manuscript. In LaTeX left quotes should be made written as ``.

In the intro, they say “Since the experimental breakthroughs [4, 5] quantum jumps have been observed in many atomic and solid-state single quantum systems see e.g. [1] for an overview.” This is an overclaim. One only observed a trajectory when the process is driven by an instrument. That is, we must observe the outcomes of seeing trajectories. Without this, the trajectories cannot be unique.

See “to permit to permit” on page 1.

See “proviso” on page 2.

Reviewer #4 (Remarks to the Author):

This article concerns the study of unravellings of non-markovian master equations.

To this end the authors introduce the notion of influence martingale. On the whole the paper is well written and I do not find any theoretical issues. Nevertheless all the computations are not surprising since unravelling master equations using martingale is a well known theory.

The paper deserves publications but not in a leading journal as nature communications.

Reply to Reviewer # 3

We thank the Reviewer for having had the patience of carefully reading our paper once more. We are in particular grateful to the Reviewer for stating

Reviewer: *I really appreciate the authors spending a great deal of time revising this manuscript and responding to my previous report in great detail.”*

We are also very pleased to read that despite certain points of disagreement the Reviewer found the paper much more approachable.

We fully agree when the Reviewer says that if

one derives a master equation from total Hamiltonian dynamics, and makes sure that the reduced state of the system is consistent with the total dynamics then that master equation is physically motivated.

Indeed this is the reason why we ultimately got interested in this problem from the beginning.

Coming to the issues raised by the Reviewer

Reviewer: *After reading their comment I am still not convinced that if one takes a model with the density matrix is always diagonal then the corresponding influence martingale will be positive*

Our point only is that in an equation of the form

$$\frac{dP_k}{dt} = \sum_{\ell \neq k} (A_{k\ell} P_\ell - A_{\ell k} P_k) = \sum_{\ell \neq k} A_{k\ell} P_\ell - R_k P_k$$

the $A_{k\ell}$'s need to be positive definite in order to permit the interpretation of a “classical” master equation such as those considered in standard books about stochastic processes as our ref [43] (Klebaner’s book). We do not imply that a non-classical master equation must be quantum either. We just say that in the presence of negative (and bounded!) $A_{k\ell}$'s one can obtain the equation above as the diagonalization of a master equation with non-positive Lindblad weights. We also would like to add here that a completely positive but not completely positive divisible dynamical map can be expressed and conveniently analyzed as a special solution of a master equation with non-positive definite Lindblad weights.

Reviewer: *Issue 1 is still important foundational but does not challenge the validity of the method. However, I do think it should be addressed in the paper a bit more formally. The current claims seem over the top; either prove that nonpositivity must be quantum or don't say it. They should soften the claim.*

We entirely re-wrote the end of the section “Interpretation”. We now make clear that our arguments have a heuristic character and no foundational aims.

Reviewer: *Issue 2 is not important, because if one derives a master equation from total Hamiltonian dynamics, and makes sure that the reduced state of the system is consistent with the total dynamics then that master equation is physically motivated. This is different from what Shaji-Sudarshan wanted to do; they were worried about quantum process tomography and NCP dynamics. This problem is better solved using the language of instruments (see Milz-Modi PRX Quantum 2021). Importantly, the authors show that their method gives the same results as the*

master equation.

In the new version of the paper we gave more emphasis to the paper of Hartmann and Strunz ref [21] which addresses first principle derivations from total Hamiltonian dynamics. We also explicitly recall that a completely positive but not completely positive divisible dynamical map can be described as a special solution of a master equation whose fundamental solution is completely bounded. Finally we introduced the reference to the PRX Quantum by Milz and Modi to be more nuanced on the matter of non-positive maps.

Reviewer:

I still don't understand the Poisson noise issue. This seems like a big assumption. Can the authors argue that this does not take away from the generality of their results? My feeling is that the wording is the issue, not the derivation. If this is indeed the case, I will recommend publication.

We do not talk anymore about (non-homogenous!) Poisson processes but now use the terminology “counting process” whose meaning and generality is perhaps more transparent. Counting processes are the widely accepted paradigmatic models of collapse as explained in the books by Breuer and Petruccione and Wiseman and Milburn. They are attuned to describe individual detection events and they are most conveniently simulated in numerical applications. Technically one can also use the chaos decomposition (see e.g. the book Di Nunno, Giulia and Øksendal, Bernt and Proske, Frank “Malliavin calculus for Levy processes with applications to finance” Springer 2009, reference not quoted in the paper) to express a larger class of stochastic processes with finite quadratic stochastic variation in terms of counting processes. For all these reasons we do not think that our working hypotheses are restrictive.

Reviewer: *Minor comments: The usage of quotations is off throughout the manuscript. In LaTeX left quotes should be made written as “.*

Thank you for bringing this to our attention, we have fixed the quotation marks.

Reviewer: *In the intro, they say “Since the experimental breakthroughs [4, 5] quantum jumps have been observed in many atomic and solid-state single quantum systems see e.g. [1] for an overview.” This is an overclaim. One only observed a trajectory when the process is driven by an instrument. That is, we must observe the outcomes of seeing trajectories. Without this, the trajectories cannot be unique.*

We have changed the sentence so that it is now explicitly mentioned that the trajectories were seen in systems under indirect measurement.

Reviewer: *See “to permit to permit” on page 1.*

We have fixed the typo.

Reviewer: *See “proviso” on page 2.*

We changed proviso for requirement.

Reply to Reviewer # 4

We thank the Reviewer for reading the paper. We are glad to read that the Reviewer has not encountered any theoretical issue in the manuscript.

Reviewer:

To this end the authors introduce the notion of influence martingale. On the whole the paper is well written and I do not find any theoretical issues. Nevertheless all the computations are not surprising since unravelling master equations using martingale is a well known theory.

In our manuscript we deal with completely bounded divisible dynamics which is a physically relevant generalization of the well known theory which applies to completely positive divisible maps.

The martingale we introduce has the function to attribute negative weights to some quantum trajectories. This is crucial in order to generate Monte Carlo averages converging to a completely bounded dynamical map. Mathematically, our martingale does not define a standard change of measure.

The role of the martingale in our work is thus fundamentally different from that in the earlier works by e.g. Barchielli and collaborators. In their studies, strictly positive martingales specify a change of measure (Girsanov's formula) a standard operation from the point of view of stochastic process theory. The purpose of the change of measure is then to express the non-linear norm-preserving stochastic evolution of the system's state vector as a linear evolution. We have now commented on this fundamental difference between our work and the earlier works on page 8 in the discussion.

Finally we wish emphasize that our findings provide a simple Markovian unraveling in the Hilbert space of the system extending the domain of applicability of quantum trajectory theory to cases not previously covered by the theory. The very existence of such Markovian unraveling and the fact that it admits a straightforward numerical implementation are in our view surprising results which represent a substantial advance of quantum trajectory theory.

REVIEWERS' COMMENTS

Reviewer #4 (Remarks to the Author):

The authors have improved their manuscript during the procedure of refereeing. The results of the paper are new and mathematically rigorous.

I am still not convinced that it is a major breakthrough in the theory of quantum trajectories. At least it gives a way to simulate time-convolutionless equation with negative coefficients but it is not clear whether it is a big challenge or not. Furthermore the part concerning non negative probabilities and the potential interpretation in terms of quantum measurement can be debated. Diffusive equations which can also be used in the context of quantum measurement are not really evoked. Why?

At this stage due to the level of impact required by Nature journals, I would probably suggest the authors to consider more specialized journals.